# LATENT FOURIER TRANSFORM

**Mason L. Wang**
CSAIL
Massachusetts Institute of Technology
ycda@csail.mit.edu

**Cheng-Zhi Anna Huang**
CSAIL
Massachusetts Institute of Technology
huangcza@mit.edu

## ABSTRACT

We introduce the Latent Fourier Transform (LATENTFT), a framework that provides novel frequency-domain controls for generative music models. LATENTFT combines a diffusion autoencoder with a latent-space Fourier transform to separate musical patterns by timescale. By masking latents in the frequency domain during training, our method yields representations that can be manipulated coherently at inference. This allows us to generate musical variations and blends from reference examples while preserving characteristics at desired timescales, which are specified as frequencies in the latent space. LATENTFT parallels the role of the equalizer in music production: while traditional equalizers operates on audible frequencies to shape timbre, LATENTFT operates on latent-space frequencies to shape musical structure. Experiments and listening tests show that LATENTFT improves condition adherence and quality compared to baselines. We also present a technique for hearing frequencies in the latent space in isolation, and show different musical attributes reside in different regions of the latent spectrum. Our results show how frequency-domain control in latent space provides an intuitive, continuous frequency axis for conditioning and blending, advancing us toward more interpretable and interactive generative music models.

## 1 INTRODUCTION

Modern audio generation models often operate in a coarse-to-fine manner, generating progressively finer representations of the output signal in a conditional chain. In diffusion models (Kong et al., 2020; Liu et al., 2023; Huang et al., 2023), higher noise levels provide coarser representations, while lower noise levels provide finer representations. In autoregressive models like AudioLM (Borsos et al., 2023a) and MusicLM (Agostinelli et al., 2023), an encoding stage represents the input signal as a hierarchy of coarse-to-fine tokens, and a generative model attempts to predict fine tokens from coarser ones. This is also the case for masked token models (Garcia et al., 2023), discrete diffusion (Yang et al., 2023), and next-scale prediction (Qiu et al., 2024).

Since the generative process involves conditioning on coarse representations, it is natural to generate new samples using the coarse representations of a reference example. This type of conditioning has been used for stroke-based image editing and image translation (Meng et al., 2021; Choi et al., 2021). However, conditioning on small- or mid-scale features is harder, since the representations used by the generative model rarely capture these features in isolation. For instance, discrete representations define fine tokens *relative* to coarse ones via residual vector quantization (RVQ) (Zeghidour et al., 2021; Kumar et al., 2023), preventing them from being interpreted independently.

Conditioning on arbitrary timescales from a reference example would be useful in music, which contains slow-moving patterns (like chord progressions) and fast-moving patterns (like trills). Patterns occurring at different timescales may be desirable starting points for generating musical variations, but are difficult to specify precisely using text. Existing reference-based controls (Villa-Renteria et al., 2025; García et al., 2025) target attributes like pitch, loudness, and instrumentation, which are distributed across multiple timescales. While these methods provide control over various semantic axes, none directly expose the 'timescale' axis.

To address this, we explore the use of the Fourier transform, which provides a decomposition of a signal into oscillations at different frequencies. High frequencies capture the most rapid variations in

the signal ('small-scale' characteristics), while low frequencies capture slow variations in the signal ('large-scale' characteristics). This representation has two benefits:

- First, frequency components are orthogonal, meaning that changing the signal's representation at one frequency does not affect the signal's representation at other frequencies. Thus, the Fourier transform provides an inductive bias for separating information across timescales.

- Second, the frequency axis provides an intuitive, continuous axis for specifying timescales precisely. The user can select for patterns based on the timescales *in Hz* at which they occur, instead of relying on heuristic approaches for timescale specification.

Our approach merges the Fourier transform with deep representation learning: we use a diffusion autoencoder (Preechakul et al., 2022) to *capture* musical patterns, and a latent-space Fourier transform to *separate* them by scale. To achieve synergy between these two components, we propose a simple end-to-end training framework: an encoder transforms audio into a time series of latent vectors, which is randomly masked in the Fourier domain. Then, a decoder attempts to use this frequency-masked latent sequence to reconstruct the audio with a diffusion-based objective.

After training, we can encode user-selected music into a sequence of latent vectors. Then, we can apply a Fourier transform to this latent sequence, creating a *latent spectrum*. The latent spectrum maps different musical patterns to different frequencies in it, which we refer to as *latent frequencies*. These latent frequencies correspond to the timescales at which the musical patterns occur. The user can *hear* different parts of the latent spectrum in isolation, or *generate* variations while conditioning on patterns at desired timescales, which are specified as latent frequencies. Separation between timescales also allows us to *blend* two musical examples together, retaining features at user-selected timescales from each. In short, we introduce novel frequency-based controls for generative models.

To explain these controls and their effects, we draw parallels between our framework and the *equalizer* (EQ), an essential tool in audio signal processing. The equalizer manipulates the *audible spectrum*, or the frequencies in the audio *waveform* within the limits of human hearing (20 – 20,000 Hz). This shapes sonic characteristics like "warmth," "brightness," "clarity," and "shine," which relate to different frequency ranges (Izhaki, 2017, pp. 223–232). The equalizer is particularly crucial for *mixing* multiple musical elements together coherently, by highlighting frequencies from each element and ensuring that elements do not "clash" over similar frequency ranges (Owsinski, 2017, pp. 14, 160–161). Since the equalizer operates on audio *waveform* frequencies, it is unable to change musical or structural patterns (like notes or chords). These are more complex than waveform oscillations, and unfold on temporal scales below 20 Hz, where such oscillations are inaudible. Still, these structural patterns are also vital to combining multiple musical elements together in a coherent way.

By operating on the *latent spectrum* instead of the *audible spectrum*, our framework provides a complement to the traditional equalizer that operates on musical patterns instead of sonic qualities. For instance, we can *blend* sounds together in musically coherent ways, while preserving patterns from each sound at user-specified latent frequencies. This is akin to the way traditional EQs are used to *mix* sounds together in musically pleasant ways, by choosing which audible frequencies of each sound to highlight. We dub our framework LATENTFT, and show several applications:

1. LATENTFT can generate musically coherent variations of a given song, while preserving patterns at desired timescales. These timescales are specified as a mask over the latent frequency spectrum. (Sec. 4.2).

2. LATENTFT can blend two songs, preserving patterns from each at desired timescales. These timescales are specified as masks over the latent frequency spectrum. (Sec. 4.3).

3. We can 'zoom-in' on parts of the latent spectrum, allowing us to *hear* musical patterns at desired timescales, which are specified as latent frequencies (Sec. 4.5).

4. We can interpret the latent spectrum of a song, and show where various musical characteristics like genre, tempo, and pitch reside on the latent spectrum (Sec. 4.6).

We demonstrate these applications through quantitative metrics (Table 1), listening tests (Sec. 4.4), and qualitative examples, which can be found on our website[1].

---

[1] https://masonlwang.com/latentfouriertransform/

## 2    RELATED WORK

**Audio Generation.**    Recent years have witnessed a great expansion in audio-domain generative models, which operate in a continuous domain or by generating discrete tokens. Diffusion models (Sohl-Dickstein et al., 2015; Ho et al., 2020; Song et al., 2020) generate samples by iteratively denoising pure Gaussian noise. Other approaches to audio generation rely on discrete audio codec tokens (Zeghidour et al., 2021; Kumar et al., 2023), which compress audio into a multi-layer sequence of tokens, with successive layers capturing increasingly fine details. Token generation can proceed in an autoregressive (Borsos et al., 2023a; Copet et al., 2023; Agostinelli et al., 2023) or non-autoregressive (Garcia et al., 2023; Borsos et al., 2023b) manner, but in both cases, coarse tokens typically condition the generation of finer ones. We propose Fourier-based representations that let us condition on features at arbitrary scales. We compare our method to conditioning on intermediate or fine tokens in our Masked Token Model baseline.

**Controls for Audio and Music Generation.**    Current audio generation methods offer global controls like text (Forsgren & Martiros, 2022; Huang et al., 2023; Liu et al., 2023; Copet et al., 2023; Agostinelli et al., 2023; Chen et al., 2024; Schneider et al., 2024; Evans et al., 2025), or time-varying controls based on musical attributes like pitch and loudness curves (Wu et al., 2024; García et al., 2025) or stems (Parker et al., 2024; Villa-Renteria et al., 2025). Different time-varying signals allow for control along different semantic axes, but not along the 'timescale' axis. These works mostly condition on the *entire* control signal, not selected frequency components. The exception is Sketch2Sound (García et al., 2025), which optionally smooths the pitch or loudness-based control signal using median filtering. Still, this type of filtering is heuristic, applies only to preserving large-scale features, and operates on hand-extracted features instead of latent ones. Guidance (Levy et al., 2023) and initial noise optimization (Novack et al., 2024) have also been used to control music generation using differentiable objectives. We use guidance for our tasks in our Guidance baseline.

**Image Editing Frameworks.**    The coarse-to-fine paradigm lends itself to image editing frameworks that generate variations of input examples based on their low-frequency features. SDEdit (Meng et al., 2021) enables stroke-based image generation and editing by adding white noise to a given reference (which acts like a heuristic low-pass filter), and running the denoising process. Similarly, Iterative Latent Variable Refinement (ILVR) (Choi et al., 2021) can generate variations of images while preserving large-scale structure. During the denoising process, ILVR continually replaces the low-frequency components of the noisy sample with the low-frequency components of a (noised) reference, enabling image translation and stroke-based editing. ILVR does not condition on high-frequency or mid-frequency components, but we attempt this in our ILVR baseline.

**Fourier-Based Deep Learning.**    While we apply the Fourier transform to latent vectors, many works use frequency-domain representations of the input or output space. These include works in vision (Lee et al., 2018; Yang & Soatto, 2020; Atzmon et al., 2024) and audio (San Roman et al., 2023; Moliner et al., 2024). Similar to our method (Sec. 3), Zheng et al. (2024b) propose a frequency-masked autoencoder that extends the masked image modeling paradigm (He et al., 2022; Xie et al., 2022) to the frequency domain. AudioMAE (Huang et al., 2022) applies masked image modeling to audio spectrograms, randomly masking time-frequency bins in the audio spectrogram domain. However, our method masks *latent-space* frequencies.

Other works *do* apply the Fourier transform to hidden states, but do so as part of black-box architectural units, and focus on downstream tasks instead of directly using the latent spectra. This use of the Fourier transform has been shown to improve learning in language (Lee-Thorp et al., 2021; He et al., 2023) and vision (Rao et al., 2021; Chi et al., 2020; Guibas et al., 2021; Lin et al., 2023).

Finally, some works apply the Fourier transform *post-hoc* to latent states of pretrained models, *choosing* and *interpreting* latent-space frequencies. PRISM (Tamkin et al., 2020) shows that different frequency bands of language model embedding sequences are useful for different downstream tasks. In vision, Khan et al. (2017) shows that the spectra of intermediate activations in a pretrained CNN can be used to categorize scenes. These works focus on *analysis*, while we focus on *synthesis*: we can isolate frequencies in the latent representation, but also *invert* them and observe their realizations in the input domain. Applying frequency-domain manipulations *post-hoc* to pretrained representations fails to *synthesize* coherent audio, which we show in the DAC and RAVE baselines and our ablations (Appendix B.1). This shortcoming motivates our frequency-masking strategy during training, which deliberately encourages our latents to be manipulable in the frequency domain.

**Blending.** LATENTFT can blend two examples together while choosing timescales from each (by selecting latent frequencies from each example). This is like style transfer in images (Ashikhmin, 2003; Gatys et al., 2016; Johnson et al., 2016; Huang & Belongie, 2017; Deng et al., 2022; Efros & Freeman, 2023), which merges "content" from one image with the "style" from another. Applying these methods to music is challenging due to the *multiscale* nature of musical style, as style can refer to "high-level compositional features" or "low-level acoustic features" (Dai et al., 2018). We ameliorate this ambiguity by introducing *frequency-based* controls, which provide a continuous axis for specifying which timescales we want from each input. In contrast, existing works in musical style transfer focus on specific aspects of music like timbre (Huang et al., 2018; Li et al., 2024; Wang et al., 2024), musical arrangement (Cífka et al., 2020), or composition (SE, 2016). Traditional techniques are also used to blend sounds, as done in the Cross Synthesis baseline (Smith, 2011).

## 3 METHOD

### 3.1 BACKGROUND

**Discrete Fourier Transform.** The discrete Fourier transform[2] (DFT) correlates an input signal $\boldsymbol{x} \in \mathbb{C}^N$ with $N$ complex sinusoidal signals, giving its spectral representation $\boldsymbol{X} \in \mathbb{C}^N$. The $k$th DFT coefficient is given by:

$$\boldsymbol{X}[k] = \boldsymbol{x} \cdot \boldsymbol{w}_k, \tag{DFT}$$

where $(\cdot)$ denotes the complex dot product, and $\boldsymbol{w}_k[n] = e^{j(2\pi k/N)n}$ denotes the $k$th complex sinusoid. The complex sinusoids $\boldsymbol{w}_1, ..., \boldsymbol{w}_N$ form an orthogonal basis for $\mathbb{C}^N$, allowing the DFT to be inverted:

$$\boldsymbol{x} = \frac{1}{N} \sum_{k=0}^{N-1} \boldsymbol{X}[k]\boldsymbol{w}_k \tag{IDFT}$$

The inverse DFT is also called the "synthesis" equation, since it expresses $\boldsymbol{x}$ as a weighted sum of complex sinusoids. To provide more concrete intuition, if $\boldsymbol{x}$ is real-valued, we can express $\boldsymbol{x}$ as the sum of *real* sinusoids with various frequencies $\frac{k}{N}$, amplitudes $A_k$, and phase shifts $\phi_k$:

$$\boldsymbol{x}[n] = \sum_{k=0}^{\lfloor N/2 \rfloor} A_k \cos\left(2\pi \frac{k}{N} n + \phi_k\right) \tag{1}$$

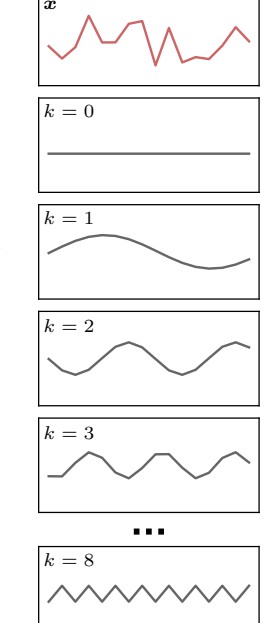

Figure 1: $\boldsymbol{x} \in \mathbb{R}^{16}$ decomposed via Eq 1.

Where $A_k$ and $\phi_k$ are both derived from the coefficient $\boldsymbol{X}[k]$, as shown in Appendix D.1. In words, the DFT can decompose a *real* signal into a sum of *real* sinusoids of different frequencies, all of which are mutually orthogonal. We show this decomposition for an example signal in Fig. 1.

**Diffusion Autoencoders.** The diffusion autoencoder was proposed by (Preechakul et al., 2022) to harness the power of diffusion models for representation learning. During training, an encoder maps an image $\boldsymbol{x}_0$ into a non-spatial semantic vector $\boldsymbol{z}_{\text{sem}}$. Then, a diffusion model (which acts as the decoder) tries to reconstruct $\boldsymbol{x}_0$ from $\boldsymbol{z}_{\text{sem}}$ and a noisy version of the image $\boldsymbol{x}_\tau$. Diffusion autoencoders are typically trained with a MSE loss that determines how well $\boldsymbol{x}_\tau$ is denoised, (or equivalently, how well $\boldsymbol{x}_0$ is reconstructed). During inference, $\boldsymbol{z}_{\text{sem}}$ can be used to condition a generative diffusion process and produce an image.

We have three motivations for using a diffusion autoencoder. First, the decoder harnesses the generative power of a diffusion model, allowing it to generate high-quality music even when information has been removed (masked) from the latent conditioning vector. Second, since the generative process is random, one can generate multiple variations for the same input condition. Third, diffusion autoencoders have been shown to yield latent representations $\boldsymbol{z}_{\text{sem}}$ that are semantically meaningful and linear, supporting interpolation between images and attribute manipulation. In fact, recent work shows the applicability of diffusion autoencoders to music representation learning (Pasini et al., 2024; Bindi & Esling, 2024).

---

[2]We present a simplified notation for the DFT for clarity and brevity. Note that $\boldsymbol{w}_k$ is typically presented as the complex conjugate of what we have here.

Figure 2: **Latent Fourier Transform (LATENTFT).** We encode audio (which may be represented as a waveform or spectrogram) into a series of latent vectors and compute a latent spectrum. During training (red), this spectrum is masked randomly and used to reconstruct the input. During inference (blue), the user specifies a spectral mask, which selects features from the input at specific latent frequencies and conditions a generative process.

## 3.2 METHOD OVERVIEW

Our goal is two-fold. First, we want to map an audio waveform or spectrogram $x_0$ into a time series of latent vectors, whose spectrum encodes semantic patterns. We refer to the DFT spectrum of this latent time series as the *latent spectrum*. It is important to distinguish the latent spectrum from the audible spectrum: The audible spectrum refers to the DFT spectrum of the audio waveform, and captures variations in the waveform occurring at different frequencies. In contrast, the latent spectrum captures variations in the latent time series occurring at different frequencies, which we correspond to musical patterns occurring at different timescales. Second, we should be able isolate features at selected latent frequencies and use them to *generate variations*, blend them with other audio clips, or hear them in isolation. These goals motivate an end-to-end *encoder-decoder* architecture that *encodes* music into latent spectra, and *decodes* latent spectra into music. We apply a latent Fourier transform and frequency-masking *during training*, shown in Alg. 1 and Fig. 2.

## 3.3 ENCODING THE LATENT SPECTRUM

**Encoder.** An encoder maps input music $x_0 \in \mathbb{R}^{C \times T}$ to a time series of latent vectors $z \in \mathbb{R}^{C' \times T'}$:

$$z = \text{Enc}_\phi(x_0) \tag{2}$$

Here, $C$ and $C'$ are the number of input and latent channels, while $T$ and $T'$ are the number of input and latent timesteps. Although $T$ and $T'$ do not have to be equal, $z$ must have a linear temporal axis in order to produce a latent spectrum. This favors convolutional architectures or networks like the U-Net (Ronneberger et al., 2015), whose skip connections promote input-output alignment. We define $f_r$ as the latent frame rate in Hz, or the number of latent vectors (frames) needed to represent one second of audio.

---

**Algorithm 1** Training.

**Input:** Audio Waveform or Spectrogram $x_0$
1: $z \leftarrow \text{Enc}_\phi(x_0)$
2: $Z \leftarrow \text{DFT}(z)$
3: $\eta \sim \mathcal{N}(0, 1)$         ▷ Sample threshold
4: $s \sim \mathcal{N}(\mathbf{0}, \Sigma)$    ▷ Sample frequency bin scores
5: $M \leftarrow \mathbf{1}_{s > \eta}$           ▷ Get Mask
6: $Z^{\text{masked}} \leftarrow Z \odot M$
7: $z^{\text{masked}} \leftarrow \text{IDFT}(Z^{\text{masked}})$
8: Sample noise level $\tau \sim p(\tau)$
9: $x_\tau \leftarrow \text{DiffusionForward}(x_0, \tau)$   ▷ Add noise
10: $\hat{x}_0 \leftarrow \text{Dec}_\theta(z^{\text{masked}}, x_\tau, \tau)$   ▷ Reconstruct $x_0$
11: $\ell \leftarrow \mathcal{L}(\hat{x}_0, x_0)$
12: Update parameters $\phi, \theta$ using $\nabla_{\phi,\theta}\ell$

---

**Latent Fourier Transform.** The *latent spectrum*[3] refers to the DFT of the latent timeseries $z$, applied to each channel in the latent timeseries:

$$Z = \text{DFT}(z), \quad Z \in \mathbb{C}^{C' \times K} \tag{3}$$

Applying the DFT along the time axis of our latent sequence represents each latent channel as a sum of $K = \lfloor T'/2 \rfloor + 1$ sinusoids (see Eq. 1). The sinusoids have $K$ different linearly-spaced frequencies, which capture variations in each latent channel at different temporal rates. The $k$th sinusoid completes $k$ cycles in $T'$ latent timeframes (see Fig. 1). For instance, the 0th sinusoid is constant, the 1st sinusoid has a period of $T'$ latent frames, and the 2nd sinusoid has a period of $T'/2$ latent frames. The sinusoids are also orthogonal from one another, creating an inductive bias for separating information across timescales.

---

[3]The DFT is different from the Short-time Fourier Transform, which yields a *spectrogram*, not a *spectrum*.

Specifically, $\mathrm{DFT}(z)$ stores $K$ complex coefficients indicating the amplitude and phase of each sinusoid along a length-$K$ frequency axis. We refer to the frequency-axis of $\mathrm{DFT}(z)$ as the *latent frequency axis*, and we call points along this axis *latent frequencies*. Like audible frequencies, latent frequencies are described in Hz. However, 1 Hz on the latent spectrum corresponds to oscillations in the *latent sequence* occurring at 1 cycle per second, instead of oscillations in the audio waveform. The $k$th sinusoid has a period of $T'/k$ latent frames or $T'/(kf_r)$ seconds, and thus a latent frequency of $f_k = kf_r/T'$ Hz.

**Increasing Spectral Granularity.**    In practice, we zero-pad $z$ at its end, expanding its temporal length by a factor of $L$. This increases the number of frequency bins by a factor of $\approx L$, allowing for more spectral granularity via *spectral interpolation* (Smith, 2007). This is especially useful for capturing very low-frequency patterns (below 1 cycle per $T'$ timeframes). We let $F = \lfloor LT'/2 \rfloor + 1$ be the number of spectral bins (sinusoids) after zero padding $z$.

## 3.4    FREQUENCY MASKING

At inference, we want to select specific frequencies from the latent spectrum to generate variations from them or hear them in isolation ('zoom-in' on them). This is accomplished by applying a latent spectral mask $M \in \{0, 1\}^F$, taking $Z^{\mathrm{masked}} = Z \odot M$. During inference, this mask is chosen by the user. During training, this mask is *randomized*: First, we sample a random scalar threshold $\eta \sim \mathcal{N}(0, 1)$, which helps decide the proportion of bins to be masked. Second, we sample $s \sim \mathcal{N}(\mathbf{0}_F, \Sigma)$, where $s \in \mathbb{R}^F$ assigns scores to each frequency bin. Third, we set the mask to keep bins whose score is greater than the threshold, setting $M = \mathbf{1}_{s > \eta}$.

**Random Threshold.**    Using a random threshold ensures a uniform distribution over the number of masked bins. In contrast, independently masking each frequency bin with probability $p$ corresponds to setting a fixed threshold and $\Sigma = I$. This results in a binomial distribution over the number of masked bins, which does not reflect the inference-time distribution of user-specified masks.

**Correlating Bins.**    Instead of masking each frequency bin independently, we create a "soft grouping" between nearby frequency bins by correlating their scores. This is done by multiplying uncorrelated scores $u \sim \mathcal{N}(\mathbf{0}_F, I)$ with a radial basis function matrix $K$:

$$K_{i,j} = c_i \exp\left(-\frac{|a_i - a_j|^p}{2\sigma^p}\right), \quad K \in \mathbb{R}^{F \times F}, \tag{4}$$

where $a_i = \log(f_i + \epsilon)$ is the frequency of bin $i$ mapped to a logarithmic axis, $p, \sigma$, and $\epsilon$ are hyperparameters, and $c_i$ normalizes each row of $K$ to have unit $\ell_2$ norm. Multiplying $s = Ku$ results in correlated scores between frequency bins, where the amount of correlation between two frequency bins is determined by their distance on a logarithmic axis. The covariance matrix of $s$ is $\Sigma = KK^T$. Ablations (Appendix B.1) show that correlating bin scores is key to our method's performance . Intuitively, masking frequency bins independently forms speckled masks where masked bins are often adjacent to unmasked ones. The unmasked bins provide strong local cues about nearby masked ones, reducing the model's ability to fill in contiguous regions of the latent spectrum during inference. In contrast, correlated bin scores form masks with larger contiguous regions, which combats the effect of spectral leakage and better reflects inference-time, user-specified masks.

Logarithmically scaling the frequency-axis is also key to performance (Appendix B.1). This is common in audio, exemplified by the Mel scale (Stevens et al., 1937), Constant Q-Transform (Brown, 1991), and others. More generally, structured signals from images (San Roman et al., 2023) to coastlines and mountains (Bak et al., 1987) have spectra that follow a $1/f^\alpha$ curve. Segmenting such spectra into groups of equal width along a log-frequency axis yields groups of roughly equal energy. This motivates our logarithmic scaling, where higher frequencies are more likely to form larger groups. Lastly, normalizing the rows of $K$ ensures equal marginal variance between every bin score $s_k$, so that all bins have the same marginal probability of being masked for any given threshold.

## 3.5    DECODING THE LATENT SPECTRUM

We transform $Z^{\mathrm{masked}}$ back into the time domain by applying the inverse DFT, obtaining a frequency-masked latent sequence $z^{\mathrm{masked}} = \mathrm{IDFT}(Z^{\mathrm{masked}})$. The decoder then uses $z^{\mathrm{masked}}$ to reconstruct the input $x_0$ from a noisy version of it (training), or to condition a diffusion process (inference).

**Training.** During training, we obtain a noisy version $\boldsymbol{x}_\tau$ of the input $\boldsymbol{x}_0$ through a forward diffusion process. This process samples a diffusion time $\tau \sim p(\tau)$ from a predetermined distribution and adds a $\tau$-dependent amount of noise to $\boldsymbol{x}_0$. We supply $\boldsymbol{z}^{\text{masked}}$ and $\boldsymbol{x}_\tau$ to the decoder, which gives an estimate of the clean input $\hat{\boldsymbol{x}}_0$:

$$\hat{\boldsymbol{x}}_0 \leftarrow \text{Dec}_\theta \left( \boldsymbol{z}^{\text{masked}}, \boldsymbol{x}_\tau, \tau \right) \tag{5}$$

Then, we compute a reconstruction loss $\ell = \mathcal{L}\left(\hat{\boldsymbol{x}}_0, \boldsymbol{x}_0\right)$, which is used to update the parameters $\phi, \theta$ of both the encoder and decoder. This procedure effectively trains a diffusion model, which can generate new outputs conditioned on $\boldsymbol{z}^{\text{masked}}$. While we do not require a particular diffusion framework, in practice we follow the ODE formulation in Karras et al. (2022). This framework preconditions the model inputs and outputs, uses approximately linear diffusion trajectories, and applies a second-order correction at each sampling step (omitted in Algs. 2 and 3 for clarity).

**Conditional Generation.** Our conditional generation task attempts to generate a variation of a reference song $\boldsymbol{y}$ that preserves characteristics at user-specified latent frequencies. The reference $\boldsymbol{y}$ is encoded and masked in the latent frequency domain to obtain $\boldsymbol{z}^{\text{masked}}$. The mask is user-specified, and typically selects low frequencies, high frequencies, or a band of intermediate frequencies. We use $\boldsymbol{z}^{\text{masked}}$ to condition a reverse diffusion process, which iteratively denoises pure Gaussian noise to yield a new variation.

**Blending.** Our blending task attempts to combine two musical references $\boldsymbol{y}_1, \boldsymbol{y}_2$ into a new song that preserves characteristics from each at user-specified latent frequencies. Like before, $\boldsymbol{z}_1, \boldsymbol{z}_2$ are obtained and masked in the latent frequency domain to get conditions $\boldsymbol{z}_1^{\text{masked}}$, and $\boldsymbol{z}_2^{\text{masked}}$. Here, the user specifies *two* masks specifying which latent frequencies to retain from *each* input. We obtain our blend by simulating the reverse diffusion process, at each step interpolating the derivatives induced by each condition (Alg. 3).

---

**Algorithm 2** Conditional Generation

**Input:** $\boldsymbol{z}^{\text{masked}}, \{\tau_i\}_{i=0}^N$ decreasing
1: $\boldsymbol{x} \sim \mathcal{N}(\boldsymbol{0}, \sigma_{\max}^2)$
2: **for** $i \in \{0, ..., N-1\}$ **do**
3: $\quad \hat{\boldsymbol{x}}_0 \leftarrow \text{Dec}_\theta \left( \boldsymbol{z}^{\text{masked}}, \boldsymbol{x}, \tau_i \right)$
4: $\quad \boldsymbol{d} \leftarrow (\boldsymbol{x} - \hat{\boldsymbol{x}}_0) / \sigma_i \quad \triangleright$ Deriv. of Noise Traj.
5: $\quad \boldsymbol{x} \leftarrow \boldsymbol{x} + (\tau_{i+1} - \tau_i) \boldsymbol{d}$
6: **return** $\boldsymbol{x}$

---

**Algorithm 3** Blending

**Input:** $\boldsymbol{z}_1^{\text{masked}}, \boldsymbol{z}_2^{\text{masked}}, \{\tau_i\}_{i=0}^N$, weights $\alpha, \beta$
1: $\boldsymbol{x} \sim \mathcal{N}(\boldsymbol{0}, \sigma_{\max}^2)$
2: **for** $i \in \{0, \ldots, N-1\}$ **do**
3: $\quad \hat{\boldsymbol{x}}_0^{(1)} \leftarrow \text{Dec}_\theta \left( \boldsymbol{z}_1^{\text{masked}}, \boldsymbol{x}, \tau_i \right)$
4: $\quad \hat{\boldsymbol{x}}_0^{(2)} \leftarrow \text{Dec}_\theta \left( \boldsymbol{z}_2^{\text{masked}}, \boldsymbol{x}, \tau_i \right)$
5: $\quad \boldsymbol{d}_1 \leftarrow (\boldsymbol{x} - \hat{\boldsymbol{x}}_0^{(1)})/\sigma_i$
6: $\quad \boldsymbol{d}_2 \leftarrow (\boldsymbol{x} - \hat{\boldsymbol{x}}_0^{(2)})/\sigma_i$
7: $\quad \boldsymbol{d} \leftarrow \alpha \boldsymbol{d}_1 + \beta \boldsymbol{d}_2$
8: $\quad \boldsymbol{x} \leftarrow \boldsymbol{x} + (\tau_{i+1} - \tau_i) \boldsymbol{d}$
9: **return** $\boldsymbol{x}$

---

## 4 EXPERIMENTS

### 4.1 EXPERIMENTAL SETUP

**Datasets and Training.** We train three versions of LATENTFT with three different encoders. We use UNet and MLP encoders with a mel-spectrogram frontend, as well as a raw audio encoder that utilizes a Descript Audio Codec (DAC) (Kumar et al., 2023) frontend. These encoders are described further and compared in Appendix A.1. Each model generates mel-spectrograms, which are inverted using the BigVGAN neural vocoder (Lee et al., 2022). We train our model on MTG-Jamendo (Bogdanov et al., 2019), a large-scale collection of over 55,000 songs spanning diverse musical genres (described more in Appendix A.5) segmented into 5.9-second musical clips. Hyperparameters for the decoders and training are in Appendix A.2 and A.3, respectively.

**Baselines.** We compare LATENTFT to various traditional and learned methods of generating or representing audio. First, we try several *generation* baselines, adapting them to our task:

- **Masked Token Model** (Garcia et al., 2023). We use the Vampnet masked token model, which generates discrete acoustic tokens from coarse-to-fine. Vampnet is trained to predict all acoustic tokens given a random subset of them, and supports supplying arbitrary token masks during inference. For conditional generation, we select different contiguous subsets of RVQ layers to condition on, and for the blending task, we select a different layer to take from each reference.

- **Guidance** (Levy et al., 2023). We generate mel-spectrograms with an unconditional diffusion model. At each denoising step, we compute the DFT along the time axis of the reference

spectrogram(s) and the current reconstruction $\hat{x}_0$. We compute the loss between these DFT spectra *within* the selected frequency bins, using it to update the intermediate output.

- **ILVR** (Choi et al., 2021). We generate mel-spectrograms from an unconditional diffusion model. At each denoising step, we compute DFT spectra of the intermediate output and the reference(s) set to the current noise level. We replace selected DFT frequencies of the intermediate output with the corresponding DFT frequencies of the noisy reference(s).

- **Cross Synthesis** (Smith, 2011). Cross synthesis blends two sounds by replacing the spectral envelope of one sound with that of the other. We follow the implementation in Smith (2011).

In the Guidance and ILVR baselines, note that we use the spectrum of the mel-spectrogram to steer the diffusion process instead of the latent spectrum. We also attempt *post-hoc* frequency-domain filtering of existing *representations* of audio for our tasks, similar to Tamkin et al. (2020):

- **DAC** (Kumar et al., 2023). We encode our reference(s) using Descript Audio Codec, a popular deep neural audio codec. We frequency-mask the latent states post-quantization, and feed the filtered latent sequence to the decoder to produce audio.

- **RAVE** (Caillon & Esling, 2021) offers another latent representation of the audio signal, which is often manipulated in the latent space and used to generate audio (Nabi et al., 2024; Zheng et al., 2024a). Similar to DAC, we frequency-mask the latent states obtained from the RAVE encoder, then provide them to the decoder.

- **Spectrogram.** We filter the input mel-spectrogram representation(s) directly, by computing the DFT of the mel-spectrogram(s) along the time axis, then masking the DFT(s). We convert the filtered mel-spectrograms to audio with BigVGAN (Lee et al., 2022).

In each case, we blend by taking selected frequency components from two latent representations derived from two inputs, by adding the two frequency-masked latents together before decoding.

## 4.2 Conditional Generation

We show that LATENTFT can generate variations of a given song while preserving patterns at user-specified timescales. We take 1024 random 5.9-second clips from the MTG-Jamendo test set, ensuring each clip originates from a unique song (results on more datasets in Appendix B.2). We then generate variations of each clip, conditioning on 14 different latent frequency bands of varying widths and locations (see Appendix A.6). Good variations should *adhere* to the condition, preserving input characteristics at the specified timescales, and have musically coherent, high-*quality* audio.

**Metrics.** To measure adherence, we extract time-series descriptor signals (e.g., loudness curves) from both the input and generated audio. We bandpass these descriptor signals to the selected frequency band, and measure their similarity or error with standard metrics. We select four perceptually relevant time series descriptors. First, we extract *loudness* curves following Morrison et al. (2024), and quantify their similarity using their correlation coefficient (Kosta et al., 2016). Second, we quantify *rhythmic* preservation by computing onset strength envelopes (Böck & Widmer, 2013) and measuring their beat-spectral cosine similarity (Foote et al., 2002). Third, to quantify *timbral* preservation, we extract Mel-Frequency Cepstral Coefficients and compute Mel-Cepstral Distortion (Kominek et al., 2008). Fourth, we measure *harmonic* characteristics (relating to chords and music notes) by computing tonal centroid features, and quantify error using Tonnetz distance (Milne & Holland, 2016). We measure audio quality by computing the Frechet Audio Distance (Kilgour et al., 2018) between the set of generated music and the MTG-Jamendo validation set.

**Results and Analysis.** We recommend listening to the qualitative results, which are available on the website[4], and show our variations are diverse and musically interesting. Quantitative results are in Table 1. Our model outperforms all baselines in terms of adherence, indicating that the latent spectrum captures and reproduces variations in loudness, rhythm, timbre, and harmony occurring at selected timescales. We also surpass all baselines in terms of quality. Our metrics confirm that (1) previous audio *generation* models cannot condition on features from arbitrary timescales, and (2) previous *representations* of audio are not robust to post-hoc spectral modifications.

---

[4]https://masonlwang.com/latentfouriertransform/

| | Conditional Generation | | | | | Blending | | | | |
|---|---|---|---|---|---|---|---|---|---|---|
| | Adherence | | | | Quality | Adherence to Both Inputs | | | | Quality |
| | Loud. ↑ | Rhyth. ↑ | Timb. ↓ | Harm. ↓ | FAD ↓ | Loud. ↑ | Rhyth. ↑ | Timb. ↓ | Harm. ↓ | FAD ↓ |
| Masked Tok. | - | - | - | - | 4.317 | - | - | - | - | 6.033 |
| Guidance | 0.529 | 0.813 | 1.430 | 0.099 | 1.061 | 0.557 | 0.832 | 1.607 | 0.114 | 1.466 |
| ILVR | 0.575 | 0.839 | 0.781 | 0.100 | 1.537 | 0.624 | 0.858 | **0.825** | 0.112 | 2.696 |
| DAC | 0.661 | 0.838 | 4.064 | 0.209 | 7.016 | 0.550 | 0.792 | 3.980 | 0.236 | 6.257 |
| RAVE | -0.016 | 0.718 | 3.836 | 0.180 | 4.695 | -0.006 | 0.697 | 4.439 | 0.171 | 4.478 |
| Spectrogram | 0.366 | 0.858 | 2.104 | 0.139 | 7.608 | 0.272 | 0.824 | 2.975 | 0.128 | 7.021 |
| Cross Synth. | - | - | - | - | - | - | - | - | - | 2.447 |
| LATENTFT-MLP | 0.815 | 0.963 | **0.376** | **0.079** | **0.337** | 0.686 | 0.873 | 1.021 | **0.108** | 1.387 |
| LATENTFT-UNet | 0.834 | **0.966** | 0.391 | **0.079** | 0.348 | 0.686 | **0.878** | 1.118 | 0.109 | **1.357** |
| LATENTFT-DAC | **0.878** | 0.922 | 1.390 | 0.107 | 0.915 | **0.699** | 0.846 | 1.865 | 0.131 | 1.364 |

Table 1: Results on Conditional Generation and Blending on the MTG-Jamendo Test set. Mel-Cepstral Distortion (Timbre) is divided by 100. Compared to baselines, LATENTFT variants achieve superior adherence and audio quality. The Masked Token Model and Cross Synthesis baselines do not offer frequency-based controls, so we do not compute adherence. Cross Synthesis also only applies to the blending task.

### 4.3 BLENDING

**Setup.** We show that LATENTFT can blend two songs together, while preserving patterns from each at user-specified latent frequencies. This application is motivated by the traditional equalizer, whose primary use is to promote coherence between tracks by emphasizing different audible frequencies from each of them. The experimental setup is similar to the conditional generation experiment. However, instead of selecting a single latent frequency band from a single song, we select two non-overlapping bands from two songs (details in Appendix A.6). We then measure the blended song's adherence to each song with respect to its selected subband, and average the two. To ensure that the blending is successful and musically coherent, we also report the FAD.

**Analysis.** We provide examples of blending on the website, and quantitative results are shown in Table 1. The blending task requires an adherence–quality tradeoff, since adhering to both conditions perfectly may not result in pleasant audio. Since ILVR iteratively *replaces* frequency components of the output with those of the conditions, it has a slightly better adherence score on the timbre metric, while being worse in terms of quality. ILVR also loses to LATENTFT in user studies by a substantial margin (Fig. 3) in terms of both audio quality and ability to blend. In general, LATENTFT can better adhere to two conditions simultaneously compared to baselines, and generates higher-quality audio. The ability to adhere to disjoint latent-frequency components from two reference examples also indicates that the latent spectrum separates information by timescale to some extent.

### 4.4 LISTENING STUDY

To validate our method against human preferences, we conduct a listening study comparing LATENTFT and three other systems on the blending task. We choose a discrete method (the Masked Token Model baseline), a diffusion-based method (ILVR), and a traditional method (Cross Synthesis) to compare with LATENTFT. We recruited 29 musicians to complete a 12-question survey comparing every ordered pair of systems. For each question, participants first heard two randomly-selected music clips from the MTG-Jamendo test set. They then heard two blendings of the music clips, each produced by a different system. Participants rated which blending they preferred in terms of (i) audio quality and (ii) how well the clips were merged, using two separate 5-point Likert scales. Fig. 3 shows that our model outperforms the baselines on both metrics. Additional details about the listening study and statistical analyses of the results can be found in Appendix A.7.

### 4.5 HEARING IN LATENT FREQUENCIES IN ISOLATION

LATENTFT can 'zoom in' or 'boost' patterns at specific latent frequencies, analogous to how audio engineers boost various audible frequencies to identify interesting or problematic regions (Izhaki, 2017, p. 265). We show this in Fig. 4. The first spectrogram shows an electronic music clip, containing patterns at various timescales. The second spectrogram boosts latent frequencies between 0 and 1 Hz, which removes rapid drum patterns (vertical lines near the top of the spectrograms) and

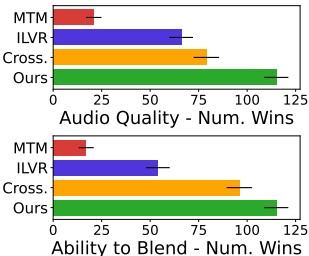

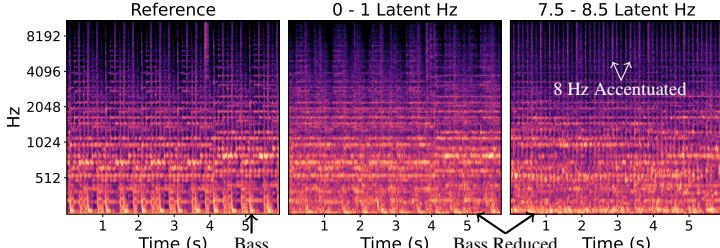

Figure 3: Listening study with pairwise comparisons. We achieve the most head-to-head wins on both criteria.

Figure 4: Isolating frequencies from an electronic music clip. We show three audio spectrograms. The second spectrogram smooths the reference spectrogram, and the third accentuates patterns occurring at 8 Hz while removing lower-frequency patterns, like the bass.

bass patterns (near the bottom), and makes the spectrogram notably smoother along the horizontal (time) axis. The third spectrogram boosts latent frequencies between 7.5 and 8.5 Hz. This accentuates a pattern in the original song occurring at 8 Hz, seen by comparing the vertical lines in the third spectrogram with those in the first. Also, the third spectrogram *does not* retain the rhythmic patterns of the bass, which occur *below* 7.5 Hz. This can be seen by comparing the lower regions of spectrograms one and three. LATENTFT allows for performing low-pass and high-pass operations on music representations *while retaining musical coherence*. Low-passing or high-passing spectrograms directly along their time axes cannot do this (Table 1). We achieve isolation using a self-blending procedure described in Appendix A.8.

## 4.6 INTERPRETING THE LATENT SPECTRUM

Musical concepts like genre, tempo, pitch, and chord changes are distributed across different regions of a song's latent spectrum, analogous to how different sonic characteristics occupy distinct ranges of the audible spectrum. Given a song, we generate many variants while performing a sweep through the frequencies we condition on. For each variant, we measure preservation of genre (using a classifier), chord progression, predominant pitch, and tempo, with respect to the original song. We plot how well the variation preserves these traits against the frequency we condition on, applying smoothing. Fig. 5 shows these traits are distributed across the latent spectrum differently. Genre is a more global feature; chords change at latent frequencies below 1 Hz; and predominant pitch and tempo reside at higher frequencies, tending to be multiples of the song's BPM. For this experiment, we use the GTZAN (Tzanetakis & Cook, 2002) dataset, since it contains ground-truth genre labels. More details about how these preservation curves computed are in Appendix A.9. Also, we interpret the latent spectra of more songs of various styles in Appendix B.3.

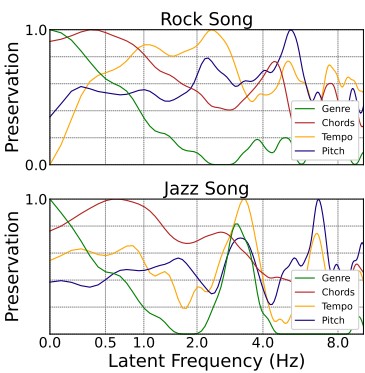

Figure 5: Preservation curves indicating where tempo, pitch, genre reside in in the latent spectra of two reference songs.

## 5 CONCLUSION

In this work, we introduced the Latent Fourier Transform, which provides novel frequency-based controls for generative models. We showed applications in conditional generation and blending in the domain of music. Future work should include enabling real-time interactivity, or disentangling the latent spectrum along semantic axes, combining both timescale-based and semantic controls.

## REPRODUCIBILITY STATEMENT

To promote reproducibility, we include code[5] for training, generating, and blending examples from LATENTFT. We also include all our baseline implementations and our experiment pipeline for the conditional generation task, the blending task, code for generating sweeps for the interpretability experiment (Sec. 4.6), and code for isolating frequency components (Sec. 4.5). We also include all our model architectures, training configurations and hyperparameters (Appendix A), and code for replicating the model architecture. Finally, we include code for preprocessing the datasets we used.

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

APPENDIX TABLE OF CONTENTS

## A    EXPERIMENTAL DETAILS

Below, we describe our experiments in more detail. We provide code for training and evaluating LATENTFT in our public github repository[6].

### A.1    ENCODERS

We experiment with three encoders:

1.  **MLP Encoder.** The audio is converted into an $80 \times 512$ mel-spectrogram. Each $(80 \times 1)$ timeframe is passed through an MLP to obtain an $80 \times 512$ latent sequence. Since each timeframe is processed independently, this encoder enforces input-output alignment, and results in no leakage between timeframes.

2.  **1D U-Net Encoder.** The audio is first converted into an $80 \times 512$ mel-spectrogram. This is processed by a 1D U-Net (Ronneberger et al., 2015; Stoller et al., 2018) with convolutions along the temporal axis to obtain an $80 \times 512$ latent sequence. While this encoder does not entirely prevent leakage between frames, the U-Net's skip-connections promote input-output alignment, allowing the encoding to be interpreted as a temporal sequence.

3.  **DAC Encoder.** We use the encoder from the pretrained Descript Audio Codec (Kumar et al., 2023) model to extract $1024 \times 512$ embeddings from the raw audio waveform. We then pass these embeddings to a 1D U-Net encoder that is identical to the one described above, except for the first convolutional layer, which is expanded to have $1024$ input channels instead of $80$ to accommodate the number of latent channels in DAC.

We find qualitatively that the U-Net encoder produces more pleasant-sounding audio when listening to latent frequencies in isolation. We also find that the U-Net encoder is better for blending, while the MLP Encoder is better for conditional generation. The DAC encoder's waveform frontend requires significantly more GPU memory, which required reducing the batch size from 256 to 64 during training. We observe in Table 1 that it is better at preserving loudness curves. Below, we provide more hyperparameters for each of our encoders.

### A.1.1    MLP ENCODER ARCHITECTURE

Our MLP encoder takes in an $80 \times 512$ mel-spectrogram, but processes each of the 512 latent timeframes independently, operating only on the channel axis. It can also be thought of as a convolutional network, where the convolutions apply to the time axis and have a kernel size of 1. It consists of a series of linear layers with SiLU activations (Hendrycks, 2016), group normalization layers (Wu & He, 2018), and residual connections (He et al., 2016). The hyperparameters for our MLP encoder are listed in Table 2.

| Attribute | Value |
|---|---|
| Input | $80 \times 512$ mel-spectrogram |
| Output | $80 \times 512$ latent sequence |
| Architecture | Frame-wise MLP |
| Hidden Dim. | 512 |
| Num. Hidden Layers | 16 |

Table 2: MLP Encoder Architecture

---

[6]https://github.com/maswang32/latentfouriertransform/

### A.1.2    1D U-Net Encoder Architecture

Our 1D U-Net encoder is a 1D version of the encoder used in Karras et al. (2022). The convolutions occur along the temporal axis. The U-Net consists of several blocks that process information at different resolutions, which are listed below in Table 3. In addition, we add self-attention layers to blocks at particular resolutions, which are also listed below in Table 3.

| Attribute | Value |
| --- | --- |
| Input | $80 \times 512$ mel-spectrogram |
| Output | $80 \times 512$ latent sequence |
| Architecture | 1D U-Net |
| Kernel Size | 3 |
| Resolutions | [512, 256, 128, 64, 32, 16] |
| Channels Per Resolution | [512, 512, 512, 768, 768, 1024] |
| Resolutions with Attention | [64, 32, 16] |

Table 3: 1D U-Net Encoder Hyperparameters

### A.1.3    DAC Encoder Architecture

The DAC encoder takes in a raw audio waveform which is resampled to 44.1 kHz. First, it creates a $1024 \times 512$ sequence of continuous embeddings using the encoder of Descript Audio Codec (Kumar et al., 2023). This sequence is passed to a 1D U-Net identical to the one in Table 3, except for the first convolutional layer, which has 1024 input channels instead of 80, to match the latent dimension of DAC. Table 4 provides details.

| Attribute | Value |
| --- | --- |
| Architecture | DAC + 1D U-Net |
| DAC Encoder Input | $1 \times 262144$ audio waveform |
| DAC Encoder Output | $1024 \times 512$ DAC embedding |
| 1D-UNet Input | $1024 \times 512$ DAC embedding |
| 1D-UNet Output | $80 \times 512$ latent sequence |

Table 4: DAC Encoder Hyperparameters

### A.2    Decoders/Diffusion Model Architecture

Our decoder (diffusion models) are 1D-U-Nets that combine convolutional layers with self-attention layers. The decoder is very similar to the 1D-UNet encoder described in Appendix A.1.2. The main difference is the decoder's input is a noisy mel-spectrogram $x_\tau$, as well as the masked latent $z_{\text{masked}}$. These two inputs are concatenated channel-wise before being fed to the U-Net. The U-Net predicts a linear combination of the added noise and the clean input $x_0$, as described in Karras et al. (2022). Again, we follow the architectures in Karras et al. (2022) and provide our code in our github repository. Details are shown in Table 5.

| Attribute | Value |
| --- | --- |
| Input 1 | $80 \times 512$ noisy mel-spectrogram |
| Input 2 | $80 \times 512$ frequency-masked latent |
| Output | $80 \times 512$ clean mel-spectrogram |
| Architecture | 1D U-Net |
| Kernel Size | 3 |
| Resolutions | [512, 256, 128, 64, 32, 16] |
| Channels Per Resolution | [512, 512, 512, 768, 768, 1024] |
| Resolutions with Attention | [64, 32, 16] |

Table 5: Decoder Hyperparameters

## A.3 TRAINING DETAILS

For the experiments in the main paper, we train for 700k iterations on 4 L40S GPUs with a logical batch size of 1024. We use a linear warmup for the first 4,000 training steps, and we apply cosine annealing to the learning rate after 350k iterations. In addition, following Karras et al. (2022), we store an exponential moving average of the model weights, which we use for inference. Hyperparameters for this are shown in Table 6. For the ablation experiments shown in Appendix B.1, we train for 350k iterations, and do not perform annealing.

| | Attribute | Value |
|---|---|---|
| **Training Schedule** | Num. Total Iters | 700k |
| | Num. Warmup Iters | 4k |
| | Num. Decay Iters | 350k |
| | Decay Schedule | Cosine |
| **Optimizer** | Optimizer | Adam |
| | Learning Rate | 1e -4 |
| | $\beta_1$ | 0.9 |
| | $\beta_2$ | 0.999 |
| **Batching** | Batch Size (Logical) | 1024 |
| | Batch Size (Per-GPU) | 256 |
| | Distribution Strategy | DDP |
| **Other** | Precision | Mixed FP32 + BF16 |
| | Grad Clip Value | 1.0 |
| | EMA Decay | 0.999 |

Table 6: Training Hyperparameters

## A.4 OTHER HYPERPARAMETERS

Here, we list the values of other hyperparameters mentioned in the Methods section (Sec. 3).

| | Attribute | Value |
|---|---|---|
| **DFT / Frequency Mask** | $L$ | 2 |
| | $\sigma$ | 0.5 |
| | $p$ | 2 |
| | $\epsilon$ | 1e-6 |
| **Diffusion** | $\sigma_{max}$ | 80 |
| | $\alpha$ | 0.5 |
| | $\beta$ | 0.5 |

Table 7: Other Hyperparameters. Full descriptions can be found in the Methods section (Sec. 3)

## A.5 DATASETS

Our experiments in the main paper use two datasets. All clips are resampled to a sampling rate of 22050 Hz.

1. **MTG-Jamendo.** MTG-Jamendo (Bogdanov et al., 2019) is a large-scale collection of over 55,000 spanning diverse genres, like classical, electronic, pop, and rock music. The dataset is publicly available, and is popular in tasks like neural audio compression, vocoding (Lanzendörfer et al., 2025), and music-tagging (Hasumi et al., 2025). We train our models on a dataset of 2.5 million 5.9-second clips from the MTG-Jamendo training split. The MTG-Jamendo dataset is used in the conditional generation (Sec. 4.2), blending (Sec. 4.3), listening study (Sec. 4.4), and isolation experiments (Sec. 4.5).

2. **GTZAN.** GTZAN (Tzanetakis & Cook, 2002) is a standard benchmark for genre classification, containing 1,000 30-second audio clips evenly distributed across 10 genres (blues,

classical, country, disco, hip-hop, jazz, metal, pop, reggae, and rock). We use GTZAN for the interpretability experiment (Sec. 4.6), since we require high-quality genre labels.

We show results on more datasets in Appendix B.2, where we perform the conditional generation and blending experiments on GTZAN and the Maestro dataset (Hawthorne et al., 2018).

## A.6 CONDITIONAL GENERATION AND BLENDING EXPERIMENTS

For these experiments, we partition the latent spectrum into 2 bands, 4 bands, and 8 bands. In each of the three partitionings, bands are equal-width on a logarithmic axis. For the conditional generation task, we condition each song on all 14 bands one-at-a-time, averaging results. For the blending task, we take two examples and condition on every possible unordered pair of bands inside the 4-band partition, for a total of six possible conditions.

## A.7 LISTENING STUDY DETAILS AND ANALYSIS

We used Prolific to recruit high-quality participants for our survey. All respondents self-identified as musicians, and all respondents reside in the United States. The respondents ranged in age from 20–73 years old, with an average age of 43.4 years old, and a median age of 41.5 years old.

The survey consists of 12 questions, each comparing two ordered pairs of systems on the blending task. Each question presents two reference recordings, and then presents two blendings of the reference clips from two different systems, for a total of four clips. The users are asked which recording they prefer both in terms of audio quality, and how well the clips were "blended" together. A screenshot of a question from our survey is shown in Fig. 6.

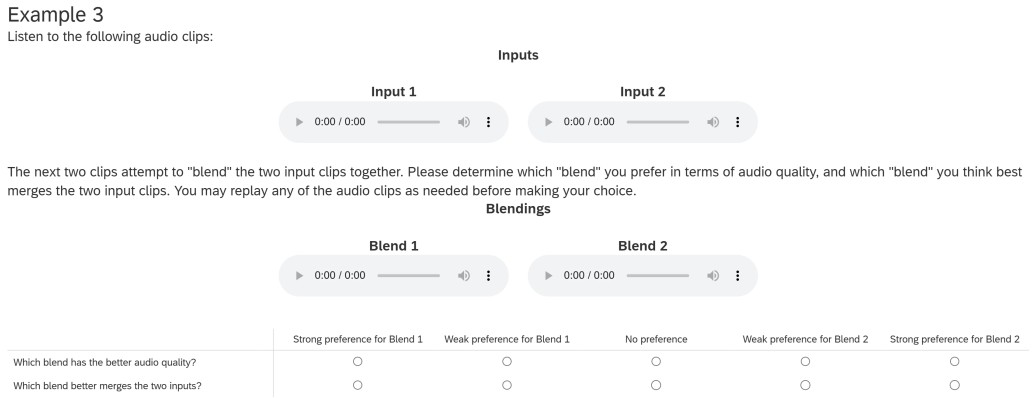

Figure 6: A question from our listening study survey. A participant will compare each ordered pair of systems in the study once.

The order of all questions is randomized. In addition, we include one attention check question for each survey participant. In the attention check, all recordings are silent, and the participant is instructed to select '2' and '4' for their two Likert scale ratings. The total duration of all the audio recordings in our survey was 5 minutes and 9 seconds. However, the median survey response time was 10 minutes and 25 seconds.

**Statistical Significance.** We performed a Kruskal-Wallis H test, which confirmed that there are statistically significant pairs among the permutations ($p = 6.4 \times 10^{-83}$). We also perform a post-hoc analysis using the Wilcoxon signed rank test. We apply a Bonferroni correction, which corrects the significant threshold to $p < 0.05/6$. According to this test, all pairs of systems have statistically significant differences in audio quality, except for the cross-synthesis and ILVR baselines. This means that LATENTFT outperforms all baselines in terms of audio quality according to our user study, to a statistically significant extent. Another Wilcoxon signed rank test indicates that all pairs of systems have statistically significant differences in "ability to blend", except for LATENTFT and the cross synthesis baseline. Pairwise significance test results are shown in Table 8.

| System 1 | System 2 | $p$-value, Audio Quality | $p$-Value, Ability to Blend |
|---|---|---|---|
| LATENTFT | Cross Synthesis | $1.59 \times 10^{-3}$ | $9.54 \times 10^{-2*}$ |
| LATENTFT | ILVR | $3.83 \times 10^{-4}$ | $8.84 \times 10^{-7}$ |
| LATENTFT | VampNet | $7.02 \times 10^{-10}$ | $1.64 \times 10^{-10}$ |
| Cross Synthesis | ILVR | $9.51 \times 10^{-2*}$ | $6.62 \times 10^{-4}$ |
| Cross Synthesis | VampNet | $1.91 \times 10^{-6}$ | $8.09 \times 10^{-10}$ |
| ILVR | VampNet | $1.55 \times 10^{-6}$ | $1.69 \times 10^{-5}$ |

Table 8: Results from a Kruskal-Wallis H test performed on listening study results. All pairs of systems have statistically significant differences in audio quality, except for ILVR and Cross Synthesis. All pairs of systems have statistically significant differences in "Ability to Blend" besides LATENTFT and Cross Synthesis. These pairs are indicated with an asterisk ($^*$).

**Inter-rater Agreement.** To compute inter-rater agreement between our 29 participants, we calculate Fleiss's Kappa, which measures the degree of agreement beyond chance for multiple raters. We report $\kappa = 0.0654$ for our question about audio quality, and $\kappa = 0.0914$ for our question about "ability to blend". Both values fall in the "slight agreement" range (Landis & Koch, 1977), indicating substantial subjective variation in perceptual judgments. This level of agreement is possible due to individual preferences and perceptual differences, which naturally lead to varied responses.

## A.8 ISOLATION EXPERIMENTS

We accomplish isolation by taking a music clip $x$ and obtaining $z$, the full-spectrum latent sequence, and $z^{\text{bp}}$, a version of the latent sequence $z$ bandpassed to the selected frequency range. We then guide the diffusion process with both $z$ and $z^{\text{bp}}$ (see Alg. 3), with blend weights $\alpha, \beta$, resulting in an output that emphasizes the selected band while suppressing content outside of it. The ratio of $\beta$ and $\alpha$ determines the amount of boosting that occurs, with $\beta \gg \alpha$ resulting in isolating the selected band almost completely.

## A.9 INTERPRETING THE LATENT SPECTRUM.

In the interpretability experiment (Sec. 4.6), we analyze the latent spectrum of individual songs, and associate different frequencies of a song's latent spectrum with musical attributes like genre, chords, tempo, and pitch. We select one song at a time to analyze. An input song is chosen from our validation split of GTZAN (Tzanetakis & Cook, 2002). We generate hundreds of variations of the input song, while conditioning on different parts of its latent spectrum. We do this by performing a linear sweep over the latent frequency axis, conditioning on every 10-bin range of the latent DFT spectrum. We measure each generation's adherence to the input song along several axes, to determine how the latent frequency that we condition on affects which attributes are preserved.

First, we classify the genre of the generated variations. We train a classifier on our training split of GTZAN, which is a linear probe on VGGish embeddings (Hershey et al., 2017), and obtains 81.8% accuracy on the validation set. Then, we apply our classifier to the generated variations, determining if the classifier's prediction of the generated variation matches the ground truth genre of the input song. For each frequency bin listed along the x-axis of our plot, we compute the accuracy across every variation whose condition included that bin. We normalize the curve to be between 0 and 1, so that it can be plotted alongside the other curves.

Second, we measure the Tonnetz correlation between the variations and the reference. This provides a proxy to measuring changes in chords, since tonal centroid features are used to identify and compute the distance between chords (Milne & Holland, 2016). Again, we plot the Tonnetz correlation between input and variation against which frequency bins we condition on. We normalize this curve to be between 0 and 1 for the sake of plotting.

Third, we measure the pitch error using the Essentia (Bogdanov et al., 2013) package. First, we use Essentia's algorithm to predict the predominant pitch (the pitch of the melody) in a song. Then, we compute the "overall accuracy" metric described in Salamon et al. (2014), to measure how well the pitches of the generated variation match with the reference. Again, we plot the pitch accuracy against

the latent frequencies that we condition on. We flip this curve vertically, so that 'up' corresponds to higher preservation instead of higher error, and normalize the curve to have minimum 0 and maximum one.

Fourth, we estimate the BPM of the variations and the reference using Librosa (McFee et al., 2015). We compute the absolute tempo error between the reference and variants, again orienting the curve so that 'up' corresponds to higher preservation, and normalizing the curve to be between 0 and 1.

We achieve the plots by applying Gaussian smoothing to all four curves. Note that unlike the blending and conditional generation experiments, we measure characteristics of the *entire* generation versus the *entire* reference, instead of bandpassing descriptor signals.

# B  ADDITIONAL EXPERIMENTS

## B.1  ABLATIONS

We ablate several components from LATENTFT-MLP to demonstrate the necessity of each component. In these experiments, we train LATENTFT-MLP and each of its variants for 350k iterations, skipping the annealing phase. Quantitative results for the conditional generation task are shown in Table 9. Quantitative results for the blending task are shown in Table 10. We also show example spectrograms for conditional generation in Fig. 9, and example spectrograms for blending in Fig. 10.

|  | Adherence | | | | Quality |
| --- | --- | --- | --- | --- | --- |
|  | Loudness ↑ | Rhythm ↑ | Timbre ↓ | Harmony ↓ | FAD ↓ |
| LATENTFT-MLP | 0.800 | **0.961** | **0.397** | **0.081** | **0.349** |
| w/o Freq. Masking | 0.476 | 0.907 | 2.675 | 0.121 | 5.341 |
| w/o Correlation | 0.694 | 0.932 | 1.284 | 0.109 | 2.744 |
| w/o Log. Scale | 0.512 | 0.838 | 1.322 | 0.097 | 1.196 |
| w/o Encoder | 0.028 | 0.565 | 3.569 | 0.130 | 0.846 |
| w/ Bandpass Augmentation | **0.861** | 0.953 | 0.562 | 0.084 | 1.511 |

Table 9: Ablation results on the **Conditional Generation** Task. Mel-Cepstral Distortion (Timbre) is divided by 100. Ablating any component of the model generally leads to worse audio quality and adherence.

|  | Adherence | | | | Quality |
| --- | --- | --- | --- | --- | --- |
|  | Loudness ↑ | Rhythm ↑ | Timbre ↓ | Harmony ↓ | FAD ↓ |
| LATENTFT-MLP | **0.678** | 0.875 | **1.030** | **0.109** | 1.371 |
| w/o Freq. Masking | 0.597 | **0.902** | 1.152 | 0.127 | 4.789 |
| w/o Correlation | 0.635 | 0.885 | 1.167 | 0.115 | 2.534 |
| w/o Log. Scale | 0.535 | 0.827 | 1.382 | 0.111 | 2.119 |
| w/o Encoder | 0.030 | 0.539 | 4.026 | 0.147 | **0.854** |
| w/ Bandpass Augmentation | 0.664 | 0.885 | 1.636 | 0.117 | 2.586 |

Table 10: Ablation results on the **Blending** Task. Mel-Cepstral Distortion (Timbre) is divided by 100. Ablating any component of the model generally leads to either significantly worse audio quality, or significantly worse adherence.

**Ablating Frequency Masking During Training.** First, we ablate frequency masking during training, applying only the inference-time user-specified mask. Previous methods apply frequency-masking post-hoc, to *analyze* a pretrained model's latent space (Tamkin et al., 2020). In Tables 9 and 10 ("w/o Freq. Masking"), we see that removing frequency-masking during training results in a substantial degradation in audio quality. Without masking during training, the decoder does not learn how to reconstruct music from frequency-masked latents, and fails to generate high-quality audio from frequency-masked latents during inference. This ablation also shows post-hoc masking is insufficient for coherent audio *synthesis*, which requires incorporating masking during training.

**Ablating Correlations between Bins.** Next, we ablate correlations between frequency bins. As explained in Sec. 3.4, we use locally correlated scores to mask frequency bins. If we mask each bin independently, we will end up with speckled, erratic masks, where unmasked bins and masked bins are next to each other. This is shown in Fig. 7. Unmasked bins provide strong local cues to nearby masked bins, making the reconstruction/denoising task easier. In contrast, masks generated from locally correlated scores are shown in Fig. 8. Our strategy of correlating scores results in large, contiguous regions of unmasked and masked bins, which makes the learning task more difficult, and better reflects inference-time, user-specified masks. Tables 9 and 10 ("w/o Correlation") verify that using an uncorrelated mask results in substantial degradations to audio quality.

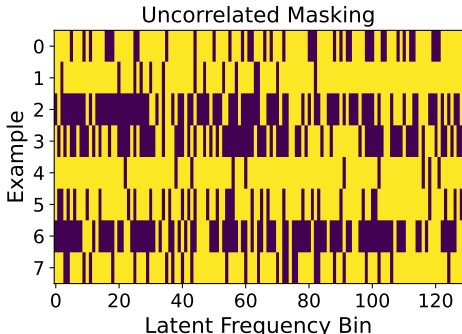 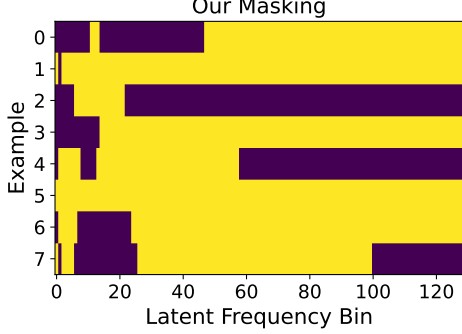

Figure 7: Example masks where there is no correlation between the scores associated with each frequency bin. The masks are speckled and erratic.

Figure 8: Example masks from our masking strategy, where bin scores are locally correlated after being mapped to a logarithmic axis. The mask forms contiguous regions.

**Ablating Logarithmic Scaling of Latent Frequency Axis.** We also ablate the logarithmic scaling of the frequency axis, discussed in Sec. 3.4. Our intuition for using a logaritmic scaling is as follows: Most structured signals have a $1/f$-spectrum, meaning that the energy at high frequencies is much lower than the energy at low frequencies. Thus, a "group" of low-frequency bins will contain much more energy than a "group" of high-frequency bins of equal width. To counterbalance this effect, we encourage high-frequency "groups" to be wider, by mapping the frequency bins to a logarithmic scale before computing correlations between bins. This reflects the fact that $1/f$-spectra have equal energy per-octave. Indeed, removing logarithmic scaling reduces both quality and adherence in both the conditional generation and blending tasks, shown in Tables 9 and 10 ("w/o Log. Scale").

**Ablating Encoder.** We also ablate the encoder, applying frequency-masking to the audio waveform directly, to show that our model's representations capture things the waveform cannot. Ablating the encoder results in poor adherence, but better audio quality in the case of blending (Tables 9 and 10 ("w/o Encoder"). Note that allowing for poor adherence improves audio quality, since the generation is less constrained. Since there is very little information in the waveform for frequencies we condition on (0–43 Hz), removing the encoder is almost like running an unconditional model.

**Random Bandpass Augmentation.** Lastly, we would like to test the necessity of using a DFT-based mask as our latent augmentation, instead of another frequency-aware latent-space augmentation. Thus, instead of applying a DFT Mask to the latent space during training and inference, we apply a randomized bandpass filter to the latent space during training, and a user-specified one during inference. We found that this resulted in some training instability, requiring several restarts. We believe the orthogonality of the DFT is helpful for training stability: From a theoretical perspective, DFT-masking in the forwards pass results in applying the same DFT mask to the upstream gradient in the backwards pass. In the backwards pass, the DFT mask can thus be interpreted as masking out orthogonal components of the upstream gradient, while leaving the unmasked components of the full-band gradient intact.

**Example Spectrograms.** Fig. 9 shows example spectrograms for the ablations on conditional generation, and Fig. 10 shows example spectrograms for blending. The figures show that many of the baselines fail to generate coherent audio. The ablation without the encoder generates coherent audio, but fails to follow the condition(s). Please refer to the figure captions for more details.

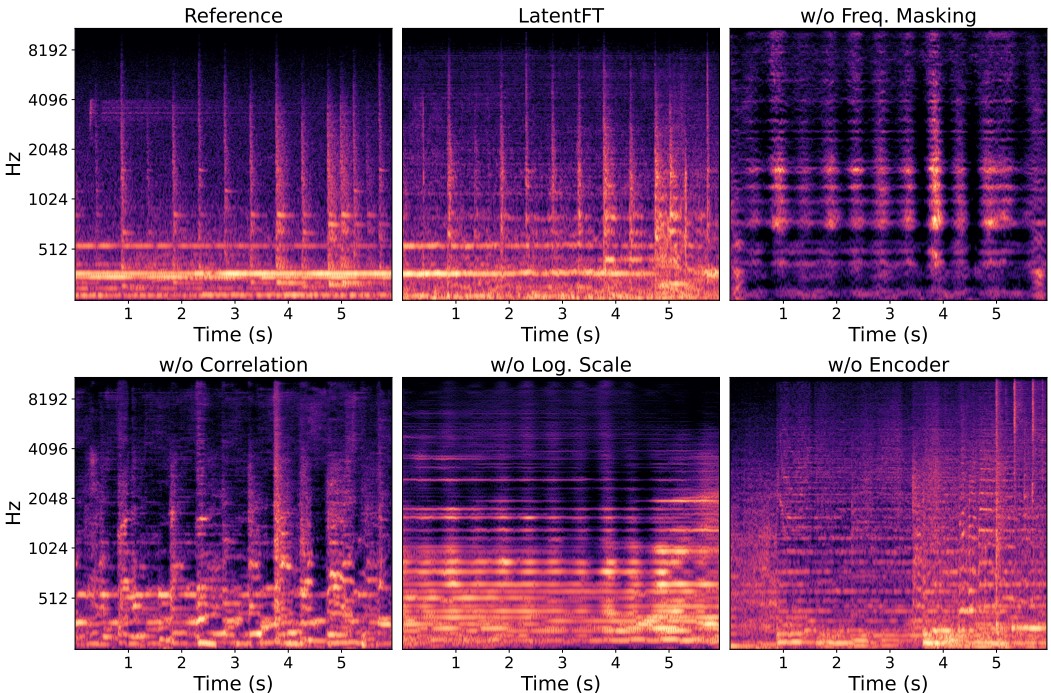

Figure 9: A conditional generation example, where we take 0.68–2.70 Hz from the latent spectrum of the reference (top left). LATENTFT generates a variation capturing the rhythmic pattern near 2 Hz. The frequency-masking, correlation, and log-scaling ablations also have a pattern near 2 Hz, but the audio quality is much worse. The encoder ablation does not follow the conditioning.

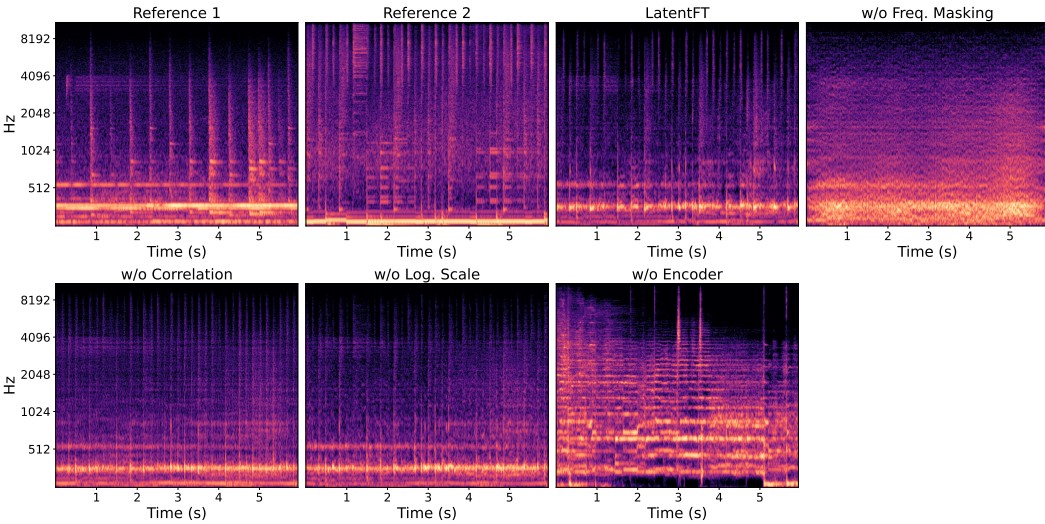

Figure 10: A blending example, where we take 0–0.68 Hz from the first reference, and 10.78–43 Hz from the second reference. LATENTFT generates a variation that contains characteristics from both examples. For instance, the rapid rhythmic patterns of Reference 2 are retained, as well as the horizontal line from Reference 1. The correlation and log-scaling ablations retain some of these characteristics, while the encoder and frequency masking ablations ignore the references.

## B.2 RESULTS ON MORE DATASETS

To demonstrate generality, we also use LATENTFT to perform conditional generation and blending on two other datasets: GTZAN and Maestro. The GTZAN dataset was previously used for the interpretability experiment in Sec. 4.6, and is described in Appendix A.5. The Maestro dataset (Hawthorne et al., 2018) is a large collection of over 200 hours of aligned piano performance audio and MIDI from the International Piano-e-Competition.

Taking our LATENTFT-MLP model trained on the MTG-Jamendo training set, we evaluate the model on 1024 5.9-second clips from both the GTZAN and Maestro datasets. The results for GTZAN are show in Table 11, and the results for Maestro are shown in Table 12. Although LATENTFT performs worse in terms of audio quality compared to our evaluations on MTG-Jamendo, we find that it outperforms our baselines on both GTZAN and Maestro. This indicates that LATENTFT can work on recordings that are only piano, or on datasets with a diverse set of genres.

| | Conditional Generation | | | | | Blending | | | | |
| | Adherence | | | | Quality | Adherence to Both Inputs | | | | Quality |
| | Loud. ↑ | Rhyth. ↑ | Timb. ↓ | Harm. ↓ | FAD ↓ | Loud. ↑ | Rhyth. ↑ | Timb. ↓ | Harm. ↓ | FAD ↓ |
|---|---|---|---|---|---|---|---|---|---|---|
| Vampnet | - | - | - | - | 5.748 | - | - | - | - | 7.173 |
| Guidance Gradients | 0.585 | 0.825 | 1.470 | 0.094 | 1.368 | 0.611 | 0.850 | 1.643 | 0.105 | 1.961 |
| ILVR | 0.628 | 0.852 | 0.730 | 0.088 | 1.873 | 0.672 | 0.877 | **0.744** | 0.097 | 3.137 |
| DAC | 0.723 | 0.845 | 4.045 | 0.191 | 8.810 | 0.610 | 0.794 | 4.115 | 0.212 | 7.162 |
| Spectrogram | 0.503 | 0.876 | 1.873 | 0.128 | 8.734 | 0.402 | 0.840 | 2.972 | 0.111 | 8.397 |
| Cross Synthesis | - | - | - | - | - | - | - | - | - | 2.884 |
| LATENTFT-MLP | 0.840 | 0.965 | **0.356** | **0.073** | **0.844** | **0.721** | 0.885 | 0.970 | **0.095** | 1.987 |
| LATENTFT-UNet | **0.855** | **0.967** | 0.377 | **0.073** | 0.905 | 0.714 | **0.891** | 1.056 | **0.095** | **1.926** |

Table 11: Results on Conditional Generation and Blending on the GTZAN dataset. Compared to baselines, LATENTFT achieves superior adherence and audio quality, demonstrating the generality of LATENTFT when it comes to new datasets with multiple genres. Mel-Cepstral Distortion (Timbre) is divided by 100. The Masked Token Model and Cross Synthesis baselines do not offer frequency-based controls, so we do not compute adherence. Cross Synthesis also only applies to the blending task.

| | Conditional Generation | | | | | Blending | | | | |
| | Adherence | | | | Quality | Adherence to Both Inputs | | | | Quality |
| | Loud. ↑ | Rhyth. ↑ | Timb. ↓ | Harm. ↓ | FAD ↓ | Loud. ↑ | Rhyth. ↑ | Timb. ↓ | Harm. ↓ | FAD ↓ |
|---|---|---|---|---|---|---|---|---|---|---|
| Vampnet | - | - | - | - | 11.914 | - | - | - | - | 14.887 |
| Guidance Gradients | 0.530 | 0.795 | 1.483 | 0.116 | 8.588 | 0.557 | 0.824 | 1.606 | 0.133 | 6.221 |
| ILVR | 0.580 | 0.817 | 0.976 | 0.118 | 9.923 | 0.627 | 0.857 | 1.007 | 0.131 | 10.018 |
| DAC | 0.729 | 0.835 | 4.088 | 0.243 | 11.745 | 0.639 | 0.776 | 3.720 | 0.297 | 11.614 |
| Spectrogram | 0.413 | 0.853 | 1.981 | 0.152 | 14.208 | 0.330 | 0.817 | 2.640 | 0.157 | 14.131 |
| Cross Synthesis | - | - | - | - | - | - | - | - | - | 3.139 |
| LATENTFT-MLP | 0.809 | 0.967 | **0.553** | **0.085** | **0.667** | 0.689 | 0.892 | **0.886** | **0.121** | 2.767 |
| LATENTFT-UNet | **0.830** | **0.968** | 0.590 | **0.085** | 0.865 | **0.710** | **0.899** | 0.943 | 0.124 | **2.708** |

Table 12: Results on Conditional Generation and Blending on the Maestro dataset. Even though the Maestro dataset is only piano recordings, LATENTFT demonstrates super audio quality and adherence compared to baselines. Mel-Cepstral Distortion (Timbre) is divided by 100. The Masked Token Model and Cross Synthesis baselines do not offer frequency-based controls, so we do not compute adherence. Cross Synthesis also only applies to the blending task.

### B.3 MORE INTERPRETABILITY RESULTS

Our interpretability experiment, introduced in Sec. 4.6, attributes parts of a particular song's latent spectrum with musical characteristics like genre, chords, tempo, and pitch. In this section, we present more examples where we analyze individual songs, and plot how well conditioning on various latent frequencies in the song preserve genre, chords, tempo, and pitch. These extra plots are show in Fig. 11. Across several musical styles, we see the trend that genre tends to lie in the frequency range around 0 Hz, indicating that it is a global characteristic. Chord changes also occur at low frequencies, with peak preservation between 0.25–2 Hz. Tempo and pitch occur at higher latent frequencies, since prominent rhythmic and melodic patterns are typically more rapid than chord changes. Please refer to Appendix A.9 for implementation details.

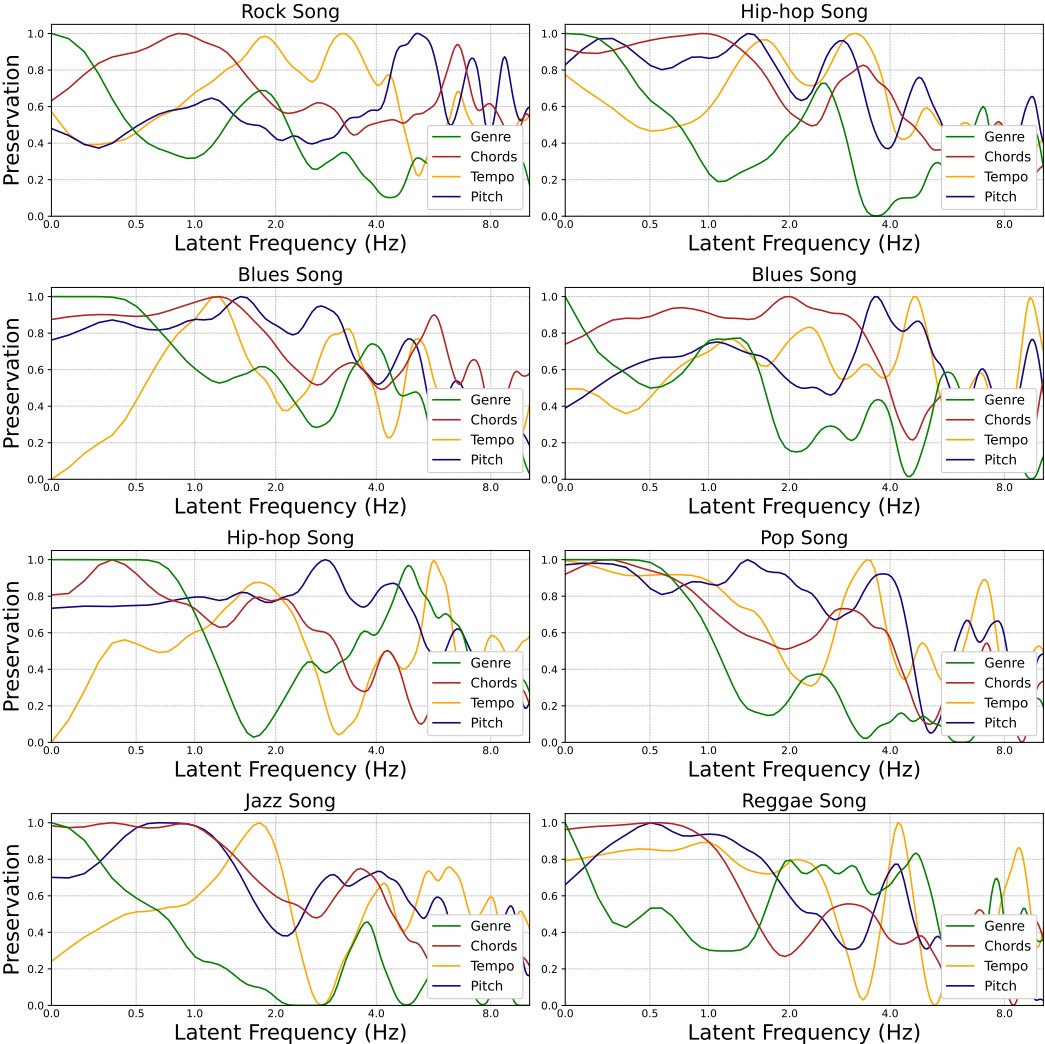

Figure 11: More Sweep Examples. Songs are taken from the GTZAN dataset. Generally, genre tends to be a global characteristic, lying around 0 Hz. Chord changes also lie in the low end of the latent spectrum, while tempo and pitch are associated with higher latent frequencies. Please refer to Sec. 4.6 for our motivations behind this experiment, and Appendix A.9 for implementation details.

### B.4   Removing the Latent DFT

In this experiment, we remove the Latent DFT entirely from both training and inference. During training, the model tries to reconstruct the input from the full latent sequence $z$. During inference, the full latent sequence $z$ remains unmasked. This is similar to the original Diffusion Autoencoder from Preechakul et al. (2022). We find that without frequency masking, the decoder reconstructs in the input without generating any interesting variations, as show in Fig. 12. For audio examples, refer to the website under "Removing DFT Masking".

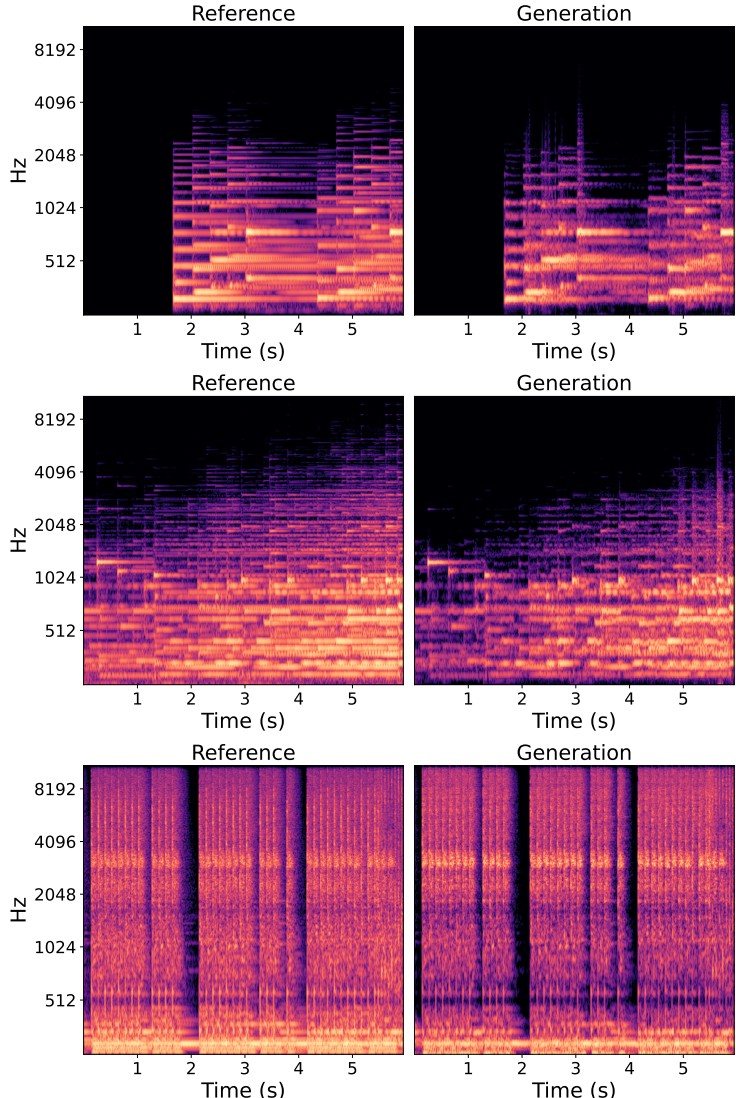

Figure 12: Mel-spectrograms where we remove the DFT during both training and inference. During inference, we condition the diffusion process on the full latent sequence $z$ derived from a reference (left). This reconstructs the input without creating a variation (right).

## B.5 PER-BAND ERROR

We show in Fig. 13 that conditioning on mid-scale or fine-scale RVQ levels leads to a rapid degradation in audio quality. On the left, we generate audio using the Masked Token Model baseline (Garcia et al., 2023), which contains 14 RVQ levels in total. We condition on each of the levels individually, and observe a degradation in quality as we condition on finer and finer tokens. On the right, we show a comparison with our model, conditioning on different latent frequency bands instead of different RVQ layers. As we condition on higher and higher frequencies (smaller timescales), the audio quality does not degrade. The metrics shown are averaged across 1024 songs from the MTG-Jamendo test set.

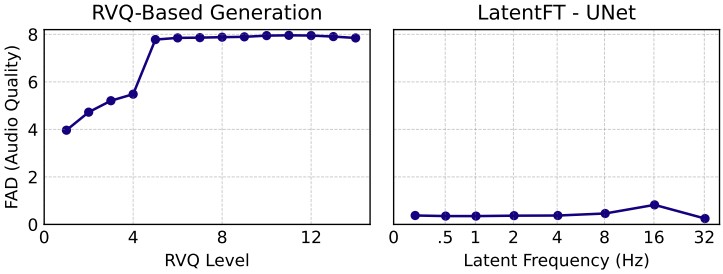

Figure 13: Conditioning on various RVQ layers in the Vampnet Model (left) and on various latent frequencies in our model (right). Our model maintains generation quality even when conditioning on finer-scale features.

## C ADDITIONAL RELATED WORK

**Separating Info by Scale.** Our work relates to learned multiscale representations that attempt to separate information by scale. Hierarchical VAEs attempt to model the data distribution using a (often multiscale) stack of latent variables. However, Zhao et al. (2017) show theoretically and experimentally that most hierarchical VAEs (Sønderby et al., 2016; Gulrajani et al., 2016; Bachman, 2016) have difficulty separating information between levels. They propose an alternative multi-network architecture under the assumption that deeper networks encode more abstract features, while shallower ones will encode simpler ones. Using these networks, they show they can vary features of an image across a few (e.g. 4) scales independently. Still, the exact scale that each network corresponds to depends on the data distribution. We extend this by 1) providing a *continuous* scale axis and 2) providing an *intuitive, non-heuristic* way of specifying scales via Hz.

**Generative Audio Equalizer.** Similar to our work, Moliner et al. (2024) introduce a diffusion-based generative audio equalizer. While this work generates content at selected *audible* frequencies, we generate content at *latent* frequencies.

**Other Uses of the Fourier Transform in Deep Learning.** The Fourier transform has also been used in CNNs to accelerate convolutions (Mathieu et al., 2013; Ding et al., 2017). Audio signals are also ubiquitously represented in the frequency domain, as are MRI images (Passigan & Ramkumar, 2024).

**AudioMAE.** Another work in audio that uses a masking strategy during training is AudioMAE (Huang et al., 2022). This work builds off of masked autoencoders for images (He et al., 2022). Here, a neural network tries to reconstruct an audio spectrogram after many time-frequency patches have been masked. This task allows the network to learn representations that are useful for classification, event detection, and retrieval. While AudioMAE masks random time-frequency bins, LATENTFT masks random bins in the latent spectrum.

## D   EXTENDED BACKGROUND

### D.1   DFT

Here, we derive Eq. 1:

$$\boldsymbol{x} = \frac{1}{N} \sum_{k=0}^{N-1} \boldsymbol{X}[k] \boldsymbol{w}_k$$

$$\boldsymbol{x} = \frac{1}{N} \sum_{k=0}^{N-1} \boldsymbol{X}[k] e^{j(2\pi k/N)n}$$

We consider the case where $N$ is odd. By the periodicity of complex sinusoids:

$$\boldsymbol{x} = \frac{1}{N} \sum_{k=-\lfloor N/2 \rfloor}^{\lfloor N/2 \rfloor} \boldsymbol{X}[k] e^{j(2\pi k/N)n}$$

We can combine the $k < 0$ terms with the $k > 0$ terms in the sum like so:

$$\boldsymbol{x} = \frac{1}{N} \sum_{k=1}^{\lfloor N/2 \rfloor} \left[ \boldsymbol{X}[k] e^{j(2\pi k/N)n} + \boldsymbol{X}[-k] e^{-j(2\pi k/N)n} \right] + \frac{\boldsymbol{X}[0]}{N}$$

Where we take the $k = 0$ out of the sum. The DFT of a real-valued signal is Hermitian, meaning that $\boldsymbol{X}[-k] = \boldsymbol{X}^*[k]$:

$$\boldsymbol{x} = \frac{1}{N} \sum_{k=1}^{\lfloor N/2 \rfloor} \left[ \boldsymbol{X}[k] e^{j(2\pi k/N)n} + \boldsymbol{X}^*[k] e^{-j(2\pi k/N)n} \right] + \frac{\boldsymbol{X}[0]}{N}$$

As its complex conjugate, $\boldsymbol{X}^*[k]$ has the same magnitude as $\boldsymbol{X}[k]$, but the opposite phase. We can split $\boldsymbol{X}[k]$ into its magnitude $|\boldsymbol{X}[k]|$ and phase $\phi_k$, and do the same for $\boldsymbol{X}^*[k]$:

$$\boldsymbol{X}[k] = |\boldsymbol{X}[k]| e^{j\phi_k}$$
$$\boldsymbol{X}^*[k] = |\boldsymbol{X}[k]| e^{-j\phi_k}$$

Plugging these into our formula:

$$\boldsymbol{x} = \frac{1}{N} \sum_{k=1}^{\lfloor N/2 \rfloor} \left[ |\boldsymbol{X}[k]| e^{j\phi_k} e^{j(2\pi k/N)n} + |\boldsymbol{X}[k]| e^{-j\phi_k} e^{-j(2\pi k/N)n} \right] + \frac{\boldsymbol{X}[0]}{N}$$

$$\boldsymbol{x} = \frac{1}{N} \sum_{k=1}^{\lfloor N/2 \rfloor} |\boldsymbol{X}[k]| \left[ e^{j((2\pi k/N)n + \phi_k)} + e^{-j((2\pi k/N)n + \phi_k)} \right] + \frac{\boldsymbol{X}[0]}{N}$$

Using Euler's Formula:

$$\boldsymbol{x}[n] = \frac{1}{N} \sum_{k=1}^{\lfloor N/2 \rfloor} 2|\boldsymbol{X}[k]| \cos\left( \frac{2\pi}{N} kn + \phi_k \right) + \frac{\boldsymbol{X}[0]}{N}$$

This can be expressed as:

$$\boldsymbol{x}[n] = \sum_{k=0}^{\lfloor N/2 \rfloor} A_k \cos\left( 2\pi \frac{k}{N} n + \phi_k \right)$$

Which is the form that we desire. Observe that the constant term outside of the sum re-enters the sum as a constant cosine ($k = 0$). The case where $N$ is even is quite similar, but includes another term (the Nyquist term), which is always a real cosine.

# E LLM USAGE

We used LLMs to help us improve the writing of our paper, for instance, by finding synonyms for certain words or for finding more concise ways to phrase particular ideas. We also used LLMs as a search tool to help us find related work, but relied on our own interpretation of the work after references were provided.

