# OpenReview forum: "Latent Fourier Transform"
_ICLR.cc/2026/Conference — ICLR 2026 Oral_

### Official Review · Reviewer_eE9f · 2025-10-27

**Soundness:** 3
**Presentation:** 3
**Contribution:** 4
**Rating:** 8
**Confidence:** 3

**Summary:**

This paper introduces Latent Fourier Transform (LATENTFT), a framework for controllable music generation that has a capability of latent frequency‑domain controls. LATENTFT is based on a diffusion autoencoder with a latent‑space discrete Fourier transform (DFT), and uses randomized frequency masking during training so that latent spectra can be manipulated coherently at inference.
LATENTFT uniquely supports conditional generation and music blending by conditioning on user‑selected latent frequency filtering.

**Strengths:**

- Unique controllabilty along with timescale and frequency on latent representation.
LATENTFT offers a continuous, interpretable frequency axis in latent space, enabling users to select which timescales to preserve or blend. This is a crisp and distinctive contribution to controllable music generation and muisc blending.

- The overall design choice is simple yet effective and well-motivated based on classical audio signal processing, e.g., DFT/iDFT with masking strategy on the basis of insights on frequency characteristics of audio.
- Empirical evidence that latent‑space frequency filtering has interesting characteristic that could pave the way to new control axis on music generation. Spesifically, int the demo page, the authors demonstrate clear difference of dynamics between singal-based and latent-frequencey-based filtering, which makes the model possible to conduct novel music blending.

**Weaknesses:**

Although other representative autoencoder-based audio syhtesis methods such as RAVE [1] and DDSP [2] do not perform explicit latent frequency filtering, the lack of disucssion and comparison with these methods remains a minor weakness. Clarifying how these approaches relate to and are positioned against LATENTFT in the main body or the appendix would make the paper stronger.

[1] Caillon, Antoine, et al. "RAVE: A variational autoencoder for fast and high-quality neural audio synthesis." arXiv preprint arXiv:2111.05011 (2021).

[2] Engel, Jesse, et al. "DDSP: Differentiable digital signal processing." ICLR 2020.

**Questions:**

- Would it be possible to benchmark RAVE and/or DDSP with the evaluation setups conducted on this paper? Alternatively, if these models are considered unsuitable for benchmarking, could the authors please elaborate on the reasons?

[Comments]
- The reviewer noice that there would be a high-level conceptual connection between the training strategy on LATENTFT and AudioMAE [3] (which is done in Mel-spectrogram domain). This would suggest that learned latent space might have strong semantic meanings not only for controllable music generation but also for audio classification and understanding tasks. Further investigation of the latent representation on such downstream tasks could provide interesting insights and would further strengthen the controbution.

[3] Huang, Po-Yao, et al. "Masked autoencoders that listen." NeurIPS 2020.

---

> ### Author Response · Authors · 2025-11-24
> **Official Comment By Authors**
>
> We thank the reviewer for their thoughtful insights and feedback. We agree that our framework presents a distinctive contribution to controllable music generation and blending by introducing a new control axis based on “timescale”. In the updated paper, changes relating to this review are in purple.
>
> ---
> **[RAVE]**
>
> RAVE provides a latent representation of audio that can also be used for generation. As a baseline, we manipulate RAVE’s latent vectors in the latent-space frequency domain, then decode them back into audio. This is a form of post-hoc frequency-domain masking, similar to our DAC baseline.
>
> We conduct an experiment that uses a pretrained RAVE model to:
> - Encode audio into a series of latent vectors,
> - Apply frequency-domain masks to these latents, and
> - Decode the masked latents back into audio using RAVE's decoder.
>
> For the **conditional generation** task, we mask the latent sequence in the frequency domain to select the latent frequencies we wish to preserve. Then, we generate a variation by decoding the frequency-masked latents, hoping that the model will preserve the unmasked portions of the latent spectrum while infilling the rest.
>
> For the **blending** task, we frequency-mask the latent sequences of two reference clips, add them together in the latent space, then decode the combined latent representation. We hope that selecting latent frequencies from each input will result in a latent representation that, when decoded, blends the inputs together while preserving selected patterns from each.
>
> We use **RAVE v2 (MusicNet)**, trained on the diverse and publicly available MusicNet dataset:
>
> - Model: https://acids-ircam.github.io/rave_models_download
> - Reference for MusicNet dataset: Thickstun et al., *Learning Features of Music from Scratch*, ICLR 2017.
>
> We have updated our paper to discuss RAVE in Sec. 4.1 (“Baselines”) and report its results on our MTG-Jamendo test set in Table 1 in the paper. In addition, we show results here, comparing RAVE to LatentFT-MLP:
>
> **Conditional Generation**
> | Method | Loud ↑ | Rhythm ↑ | Timbre ↓ | Harmony ↓ | FAD ↓ |
> |--------|--------|--------------|----------------|---------------|---------------|
> | **RAVE** | -0.016 | 0.718 | 3.836 | 0.180 | 4.695 |
> | **LatentFT-MLP** | **0.815** | **0.963** | **0.376** | **0.079** | **0.337** |
>
> **Blending**
> | Method | Loud ↑ | Rhythm ↑ | Timbre ↓ | Harmony ↓ | FAD ↓ |
> |--------|--------|--------------|----------------|---------------|---------------|
> | **RAVE** | -0.006 | 0.697 | 4.439 | 0.171 | 4.478 |
> | **LatentFT-MLP** | **0.686** | **0.873** | **1.021** | **0.108** | **1.387** |
>
>
> We observe that the RAVE baseline generates poor quality audio that fails to adhere to the input clip(s) at the timescales we would like to retain. This is indicated by the high FAD metrics, and the lack of adherence in terms of loudness, rhythm, timbre, and harmony within the selected subband. This experiment serves as further evidence that the latent spaces of current audio models are not amenable to frequency domain manipulation.
>
> ---
>
> **[DDSP]**
>
> We agree that DDSP represents a major advancement in generative audio modeling. However, we would like to point out that DDSP only works on **monophonic** audio, while our tasks are based on complex patterns within **polyphonic** music clips. DDSP can only generate monophonic audio with a single fundamental frequency. DDSP also does not expose a controllable axis corresponding to selecting **specific timescales** from a reference, making it difficult to use for our conditional generation task.  We welcome suggestions from the reviewer about potential directions at the intersection of our work and DDSP.
>
> ---
>
> **[AudioMAE]**
>
> We also agree that there is an interesting connection between our masking approach and AudioMAE. AudioMAE masks time-frequency spectrogram bins, whereas LatentFT masks bins in the latent spectrum, corresponding to different latent timescales. This results in different learning objectives:
>
> - **AudioMAE:** infilling missing spectrogram regions.
> - **LatentFT:** infilling masked portions of the latent spectrum.
>
> We have added a short discussion of this relationship in Appendix C (“AudioMAE”), and updated our related works section.

---

> > ### Comment · Reviewer_eE9f · 2025-11-25
> >
> > The reviewer appreciated the additonal experiments and discussion with other models. The reviewer decided to keep this positive score.

---

### Official Review · Reviewer_1GMu · 2025-10-31

**Soundness:** 2
**Presentation:** 1
**Contribution:** 2
**Rating:** 2
**Confidence:** 4

**Summary:**

The paper proposes controllable music generation using a Diffusion Autoencoder framework. The method involves applying a DFT over time to encoded latents (interpreted as “timescales” or “latent frequency”), random masking of selected bands, inverting back to latents via an inverse DFT, and decoding with a diffusion model. The core idea of “latent frequency” is promising and appears to offer control over musical structure including song sections, harmony, all the way down to short transient content.
The paper, however, has weaknesses in multiple fronts. The writing is unclear on core concepts and methods (e.g., what “frequency” means in latent space, what masking actually teaches the network), and lacks explanation on how to interpret fundamental quantities like latent space sampling rate. Several claims lack experimental validation and theoretical justification, and are not evaluated against adequate baselines. Although this idea is worth exploring, in its current state I unfortunately cannot recommend this paper for acceptance.

**Strengths:**

1. The model is open-sourced.

2. The engineering effort is huge.

3. Dual-NTP and SPC all considered music prior into generative modeling, solid contributions.
The model performance is competitive, considering the data they can use.

**Weaknesses:**

Timescales vs. audio frequencies: The method manipulates “frequency” in latent space, and not audible frequency bins in the waveform domain. This distinction (lines 84-89) is crucial but is not emphasized clearly enough in the methods section. Please clearly define latent frequency, latent frame rate (f_r), the mapping (f_k = k f_r/T'), and how “Hz” must be interpreted in latent space. Consider moving the (f_r) explanation from the appendix into Section 3.

It is unclear as to how well latent frequency bands align with musical structure. Please provide analyses or experiments showing whether latent bands correlate with sectional changes (say chorus/verse), beat changes, chord changes etc. In addition, it would help to show whether this mapping is consistent across diverse tracks from different styles of music. Otherwise the claims are perhaps somewhat vague.

Section 4 does not include a basic experiment that removes latent DFT (e.g., by conditioning simply on raw latents) to show why latent DFT along with random masking is necessary or what fundamental gains it offers.

Missing model architecture. Please provide the encoder/decoder architectures and key hyperparameters in the appendix. In addition, consider reducing the section on DFTs and instead add more explanations on Diffusion Autoencoders, how they are trained, loss functions and the basic motivation for choosing this architecture over say, Latent Diffusion.

Ablations section in (Appendix B.1): The text references ablations (without masking/correlation/log-scale/encoder”), but only a table is provided. No qualitative plots or thorough analyses. The authors may have inadvertently forgotten to add plots and results in this sedition. Kindly add figures and more metrics to support the takeaways.

Dataset is too narrow: All core claims are validated on short Jamendo-style clips. Add at least one other domain e.g., other genres of music, piano (MAESTRO dataset) to highlight the “latent frequency” concept and demonstrate generality. It would be interesting to see the idea extended to speech signals.

Terminology inconsistency: If authors prefer to use “timescales,” kindly define once and stick to either “timescales (latent Hz)” or just “latent frequencies.” Please do not alternate between terms as it leads to confusion.

MINOR:

In Fig. 4, please mark the vertical axis as frequency in Hz instead of bin-index. If you are logarithmically scaling the latent frequency axis, please show this clearly as well. For clarity, add a section on how to convert log-spaced spectrum back to the temporal domain via an IDFT.

Input representation ambiguity: Fig. 2 suggests STFT/mel input, while text says waveform input. Please clarify the exact input representation and update the figure accordingly..

**Questions:**

The authors provide an analogy of EQing, however I believe this can be misleading. I encourage the authors to consider other examples that illustrate “latent frequencies” from a more compositional angle i.e, emphasizing structural as opposed to spectral attributes.
How well do different latent spectrum regions map to musical structures?

Please provide experiments (beat/downbeat F-scores vs. latent frequency bands, chroma change rate vs. latent frequency bands, onset density vs. bands, ground truth chord changes) and show cross-track consistency.

What is masking teaching the network? Explain the training signal and why random contiguous band masking encourages the encoder/decoder to organize information by temporal rate, and how this yields controllable edits at test time. Can you provide experiments that show the effect of masking vs no-masking?

Can you include more audio demos on other styles of music? It is hard to guage how successful the method is given only one style of music.

**Details Of Ethics Concerns:**

None.

---

> ### Author Response · Authors · 2025-11-25
> **Official Comment by Authors (1/3)**
>
> We thank the reviewer for their insightful feedback, as it has inspired improvements to our paper. In the updated paper, changes relating to this review are in blue.
>
> ---
> **[Timescales vs. Audio Frequencies.]**
>
> We strongly agree that the distinction between latent and audible frequencies is important, and appreciate the suggestions for improved clarity. We have rewritten parts of the introduction and the methods section (see Sec. 3.3) to make this distinction more clear. We clarify our definitions, including the meaning of latent frequency (L69-71, L270-273), latent frame rate / latent space sampling rate f_r (L252-256), the mapping f_k = k f_r/T' (L274-276), and how “Hz” can be interpreted in latent space (L272-275).
>
>
> ---
> **[Structure/How well do different latent spectrum regions map to musical structures?]**
>
> We would like to further discuss the experiments in Section 4.6 (“Interpreting the Latent Spectrum”), and Figure 5. Here, we show that different musical structures like chord progressions and melody (labeled as “pitch” in Fig. 5) map to distinct ranges of the latent frequency spectrum. Chords (shown in the red line) have a stronger association with low latent frequencies, around 0.5 Hz. This is because chords change at slow temporal rates, around 0.5 Hz. On the other hand, the song’s melody (blue line) is associated with higher latent frequencies, since melodic patterns tend to be more rapid than chord changes.
>
> We conduct this experiment by choosing an input song, generating variations conditioned on various frequency ranges, then measuring how well those variations match with the input song along different axes (genre, chords, predominant pitch, tempo). For instance, we find that when we generate variations of the Jazz song (Fig. 5) conditioned on latent frequencies around 0.7 Hz, the chord progression is more likely to be preserved. The full details for how we measure preservation of chords, pitch/melody, etc. are in Appendix A.9.
>
> The mapping is not consistent between songs, but song-dependent, as attributes like “chords” and “tempo” occur at different temporal rates in different songs.
>
> **[Interpreting Latent Spectra of Diverse Styles]**
> As suggested, we expand our interpretation of various latent frequency ranges to analyze songs of more diverse genres (e.g., hip-hop, blues, pop, reggae). This is shown Fig. 11 and discussed in Appendix B.3.
>
>
> **Discussing Figure 4**
>  In addition, Fig. 4 shows visual examples of how different latent frequencies correspond to structural musical patterns. Fig. 4 shows a reference audio spectrogram, then two audio spectrograms produced when we isolate latent frequencies between 0-1 Hz and 7.5-8.5 Hz, respectively. For instance, selecting 7.5-8.5 Hz emphasizes a drum pattern near 8 Hz, while eliminating the slower-moving patterns in the bass. These observations are visible on the third spectrogram in Fig. 4.
>
> **Sectional/Beat Changes**
> We have added several examples to https://latentfouriertransform.com/#transitions (“Changes Between Sections”) to demonstrate how LatentFT handles section changes. Here, we take clips that occur at the boundary between two musical sections, and show that conditioning on low
> latent frequencies can create variations capturing the transitions between sections. We believe that the variations our model produces here result in new and interesting sectional changes.
>
> Verifying LatentFT’s ability to capture large-scale phenomena like overall form would ideally involve generating 5x-30x longer segments (30s - 3 min, instead of 5.9 seconds). Unfortunately, our computational constraints make training/generating at these lengths infeasible.
>
> ---
>
> **[Removing Latent DFT Entirely]**
>
> We perform the suggested experiment. We train a model without any DFT masking. Given a reference, we extract raw latents, and condition on these raw latents during inference, also without any DFT masking. Thus, the DFT does not play a role here, meaning that we have a typical diffusion autoencoder.
>
> We also note that without the latent DFT, the user has no way to specify what to keep from the input clip(s). This specification is performed in LatentFT by specifying a mask over the latent DFT.
>
> The model reconstructs the input with some degradations in quality, and without generating interesting variations. Audio examples can be found on the website (https://latentfouriertransform.com/#no_dft, “Removing DFT Masking”), and corresponding example spectrograms in Appendix B.4. Also, a variant where we only remove random DFT masking during training is in Appendix B.1 (L1280-1287).

---

> > ### Author Response · Authors · 2025-11-25
> > **Official Comment by Authors (2/3)**
> >
> > **[Model Architecture, Hyperparameters]**
> >
> > In addition to the existing code release and the description of model architectures in Appendix A, we elaborate on encoder/decoder architectures and training hyperparameters and descriptions in Appendix A.1-A.3.
> >
> > ---
> >
> > **[Diffusion Autoencoders]**
> >
> > We also expound upon diffusion autoencoders (training, loss functions, motivations) as requested, in Section 3.1 (“Diffusion Autoencoders”, L200-206). Since our input is a music clip and our output is also a music clip, we use an autoencoder framework.
> >
> > We choose diffusion autoencoders for three reasons:
> >
> > The diffusion decoder harnesses the generative power of a diffusion model, allowing it to generate high-quality music even when information has been removed (masked) from the latent conditioning vector.
> > Since the generative process is random, one can generate multiple variations for the same input condition.
> > Third, diffusion autoencoders have been shown to yield semantically latent representations, supporting attribute manipulation [1]. Recent work shows the usefulness of diffusion autoencoders in music representation learning [2,3].
> >
> >
> >
> > **Latent Diffusion**
> > We do not view our framework as competing with latent diffusion, but as an orthogonal matter. In fact, our framework is compatible with latent diffusion. For instance, we can use the latent-space representation of a pretrained VAE as our inputs and as our output targets (in other words, LatentFT can be nested in the latent space of a VAE).
> >
> > ---
> >
> > **[Ablations]**
> >
> > Appendix B.1 includes a discussion of our ablation studies that accompanies the quantitative metrics, which we expanded following the suggestion. Previously, these ablations were only evaluated on the blending task (Table 10). We have updated the section to report additional metrics on the conditional generation task (Table 9). As requested, we also now accompany the results in Tables 9 & 10 with qualitative plots and analysis (Figs. 9 & 10). We expand on this section to analyze the necessity of each ablated design choice, including plots (Fig. 7, 8) comparing our frequency masking strategy to ablated variants. We also include another ablation that uses an alternative latent frequency augmentation (random bandpassing), inspired by reviewer YTEd.
> >
> > ---
> >
> > **[Dataset Diversity/Audio Demos on Other Styles of Music]**
> >
> > Our experiments primarily use MTG-Jamendo, which contain many diverse genres, with 87 genre tags, like classical, orchestral, soundtrack, jazz, and world music (see https://mtg.github.io/mtg-jamendo-dataset/ for the genre distribution). Our examples on the website also include genres like classical guitar (0-0.5 Hz Example), Rock (under Blending, “Patterns are Restylized”), Christmas (under “Driving a Christmas Song Using Guitar Strumming”), and Electronic (under “Comparing the Audible and Latent Spectra”), which we chose for maximum familiarity.
> >
> > To address the reviewer’s concern, we also conduct experiments on the Maestro (piano) dataset [4] and evaluate conditional generation and blending according to our metrics (Table 12, Appendix B.2). We find that our LatentFT model trained on MTG-Jamendo can generalize, beating baselines in audio quality and adherence. Audio examples for Maestro are on the website as well (https://latentfouriertransform.com/#other_styles).
> >
> > We also evaluate on the 10-genre GTZAN dataset (results in Table 11), which includes blues, classical, and country music, among other genres [5]. These results are also in Appendix B.2.
> >
> > ---
> >
> > **[Terminology Confusion: Timescales vs. Latent Frequencies]**
> >
> > We agree we should avoid confusing or interchanging the terms “timescales” and “latent frequencies. Our initial submission uses them too interchangeably. Now, we use the term “timescale” to refer more generally to the temporal extents of particular musical patterns. We use “latent frequencies” more specifically, to refer to frequency bins in the latent DFT _within our particular framework_. We have adjusted each occurrence of “timescale” or “latent frequency” to fit these definitions. We also have adjusted the paper to avoid using “latent frequency” before it is defined and its relationship to “timescale” is made apparent (L69).

---

> ### Author Response · Authors · 2025-11-25
> **Official Comment by Authors (3/3)**
>
> **[MINOR]**
>
> **[Figure 4]**
>
> In Figure 4, we are actually showing the mel-spectrograms produced from the output audio, not a latent spectrum. This is because the patterns that we refer to are best visualized on the audio spectrogram. The frequency axis is in Hz, and represents audible frequencies. Apologies for the confusion.
>
> To clarify, the logarithmic scaling does not modify the latent spectrum itself. It is only used to compute correlations between DFT frequency bins for the purpose of generating randomized frequency masks. Specifically, the amount of correlation between two frequency bins is determined by their spacing on a logarithmic scale. No special IDFT is needed.
>
> **[Figure 2]**
>
> We thank the reviewer for the suggestion. In Figure 2, the input can be waveform (LatentFT-DAC) or spectrogram (LatentFT-MLP, LatentFT-UNet). We update this in the figure and in the writing. Although our initial experiments used mel-spectral front ends, we have since added an experiment with a raw-waveform encoder, shown in Table 1 and described in detail in Appendix A.1.3.
>
> ---
>
> **[What is Masking Teaching the Network?]**
>
> Briefly, masking the latent spectrum during training removes information about the patterns occurring at the masked latent frequencies, while encouraging the neural network to reconstruct them. During inference, we provide a masked latent spectrum from a reference track, and the neural network is able to hallucinate the missing patterns that were removed when the latent spectrum was masked.
>
> The masking itself does not organize information by temporal rate. As a CNN, the encoder naturally outputs a sequence of latent vectors along a linear temporal axis. Applying the DFT to these latent representations is what organizes the information in these vectors by timescale, along the latent frequency axis.
>
> As our ablations show (Appendix B.1), removing random frequency masking during training (but applying an inference-time frequency mask) results in poor audio quality (Tables 9 and 10). We expand the ablations section (as recommended) to clarify and discuss this more (L1280-1287).
>
>
>
> ---
>
>
> [1] Preechakul, Konpat, et al. "Diffusion autoencoders: Toward a meaningful and decodable representation." Proceedings of the IEEE/CVF conference on computer vision and pattern recognition. 2022.
>
> [2] Pasini, Marco, Stefan Lattner, and George Fazekas. "Music2latent: Consistency autoencoders for latent audio compression." International Society for Music Information Retrieval (2024).
>
> [3] Bindi, Giovanni, and Philippe Esling. "Unsupervised composable representations for audio."  International Society for Music Information Retrieval (2024).
>
> [4] Hawthorne, Curtis, et al. "Enabling factorized piano music modeling and generation with the MAESTRO dataset." International Conference on Learning Representations (2019).
>
> [5] Tzanetakis, George, and Perry Cook. "Musical genre classification of audio signals." IEEE Transactions on speech and audio processing 10.5 (2002): 293-302.

---

### Official Review · Reviewer_YTEd · 2025-11-01

**Soundness:** 3
**Presentation:** 3
**Contribution:** 2
**Rating:** 4
**Confidence:** 4

**Summary:**

This authors introduce LATENTFT, which applies Fourier transforms to latent time series from a diffusion autoencoder and trains with random, correlated log-frequency masking to enable frequency-domain control over music generation. The approach allows users to manipulate latent frequency bands at inference to generate variations preserving specific timescales or to blend references by combining disjoint frequency regions. Experiments on MTG-Jamendo demonstrate improvements over baselines in adherence metrics (loudness, rhythm, timbre, harmony) and quality (FAD), supported by a 29-musician listening study. The method offers interpretable control via "isolation" experiments linking latent Hz to musical attributes.

**Strengths:**

The authors' decision to operate in latent frequency space rather than audio frequency is well-motivated, and the correlated log-frequency masking scheme aligns naturally with musical structure and audio engineering intuitions. The proposed algorithms are clearly specified, the training procedure is straightforward, and the mask-conditioning interface provides an interpretable control mechanism. Quantitative improvements over multiple baselines and positive user study results support the core claims for the tested domain. Preservation curves and isolation experiments provide insight into what different latent frequency bands capture, and can potentially offer practitioners guidance for control.

**Weaknesses:**

(1) All experiments use 5.9-second clips from a single dataset (MTG-Jamendo) with mel-spectrogram front-ends. This raises serious questions about generalization to: (a) longer musical forms where phrase/section structure matters, (b) other audio domains (speech, non-Western music, sound design), and (c) different representation choices (raw waveform, neural codec latents). The paper's claims about "music generation" are stronger than what this limited scope can support.

(2) The most competitive baselines (ILVR, codec/spectrogram filtering) operate on spectrogram rather than latent frequency, creating an uneven comparison. I suggest the authors to train the same architecture with random band-drop augmentation in latent time but without explicit DFT structure as a baseline/ablation. This would isolate whether gains come from Fourier properties specifically or just from any structured frequency-aware regularization.

(3) While Table 3 shows that correlated/log masking matters for quality, the ablations don't clearly separate: (a) the value of training with any latent-space augmentation, (b) the specific benefit of DFT orthogonality, and (c) the role of the log-frequency parameterization. The large FAD changes suggest the masking strategy is critical, but mechanistic understanding remains limited.

(4) With only 29 participants and no reported effect sizes, confidence intervals, or inter-rater reliability, it's difficult to assess the robustness of the preference results. The study design (direct comparison on 20 pairs) is appropriate but underpowered for such strong claims.

**Questions:**

(1) Can the authors provide failure case analysis? When does latent frequency control break down (e.g., very low-Hz bands dominating, cross-band dependencies)?

(2) How does the method perform with longer contexts (30+ seconds, ideally ~3 mins) where hierarchical structure becomes important? Does the flat latent time series model capture phrase/section boundaries?

(3) What happens with alternative encoders (e.g., raw-audio encoders like Encodec)? The appendix hints that encoder architecture affects behavior; to support the broader general claim, this deserves fuller investigation.

(4) Could you add a latent-augmentation baseline mentioned above without DFT to clarify what the Fourier structure specifically contributes versus generic frequency-aware training?

(5) What are the computational costs of the increased spectral resolution (zero-padding factor L) and correlated masking during training/inference?

---

> ### Author Response · Authors · 2025-11-25
> **Official Comment by Authors (1/4)**
>
> We thank the reviewer for their insightful feedback. In the updated paper, changes relating to this review are in green.
>
> ---
> **[W1a/Q2 Longer Musical Forms, Phrase/Section boundaries]**
>
> While we have demonstrated the usefulness of LatentFT on capturing rhythmic patterns, melodies, and chord changes, the original paper did not address our framework’s usefulness on larger-scale musical phenomena like section changes and musical form.
>
> We have added several examples to https://latentfouriertransform.com/#transitions (“Changes Between Sections”) to demonstrate how LatentFT handles section changes. Here, we take clips that occur at the boundary between two musical sections, and show that conditioning on low latent frequencies can create variations capturing the transitions between sections.
>
> Verifying LatentFT’s ability to capture large-scale phenomena like musical form would ideally involve generating 5x-30x longer segments (30s - 3 min, instead of 5.9 seconds). Unfortunately, our computational constraints make training/generating at these lengths infeasible. Our architecture uses self-attention layers whose costs scale quadratically with sequence length, (our 5.9-second models take 12 days to fully train). We acknowledge that we cannot say for sure what would happen on longer clips, and that is a promising direction for future work.
>
> ---
> **[W1b Other Audio Domains]**
>
> We agree that the paper primarily focuses on music instead of other audio domains, and have rescoped our claims in the paper accordingly (see abstract). We acknowledge that our demo examples are biased towards Western music. Our evaluation dataset (MTG-Jamendo) contains many diverse genres, including approximately 4% tagged as “world” music (Also, see https://mtg.github.io/mtg-jamendo-dataset/ for more information on the genre distribution).
>
> We also now evaluate LatentFT on the Maestro dataset (piano performances) and the GTZAN dataset (10 genres). Quantitative results for this are in Appendix B.2. We also include examples from Maestro on the website at https://latentfouriertransform.com/#other_styles.
>
> ---
> **[W1c/Q3 Representation Choices]**
>
> Our paper used the mel-spectrogram as a frontend, as it reflects human perception well, and has been used commonly in audio classification [1], speech [2] and music [3]. We agree our claims could be strengthened by experimenting with a different frontend.
>
> Thus, as another experiment, we replace the mel-spectrogram frontend with a popular audio codec frontend, Descript Audio Codec (DAC) [4]. Similar to Encodec, the DAC processes raw waveforms into embeddings. The embeddings are then passed through a 1D-UNet to obtain the latent sequence $\mathbf{z}$ (see Appendix A.1 “DAC Encoder” for details).
>
> Results are reported on the conditional generation and blending tasks under “LatentFT-DAC” in Table 1. We also show them here, comparing the three LatentFT Encoders:
>
> **Conditional Generation**
> | Method | Loud ↑ | Rhythm ↑ | Timbre ↓ | Harmony ↓ | FAD ↓ |
> |---|---|---|---|---|---|
> | **LatentFT-MLP** | 0.815  |  0.963     | **0.376** | **0.079**      | **0.337** |
> | **LatentFT-UNet** | 0.834    |  **0.966**  |    0.391    |    **0.079** |   0.348      |
> | **LatentFT-DAC** |  **0.878**  | 0.922  |    1.390    |    0.107     |    0.915    |
>
>
>
> **Blending**
> | Method | Loud ↑ | Rhythm ↑ | Timbre ↓ | Harmony ↓ | FAD ↓ |
> |---|---|---|---|---|---|
> | **LatentFT-MLP** | 0.686             |      0.873     | **1.021** | **0.108** |  1.387 |
> | **LatentFT-UNet** |     0.686        |   **0.878**   |    1.118    |     0.109    |     **1.357**   |
> | **LatentFT-DAC** |      **0.699**  |     0.846      |      1.865  |     0.131     |    1.364    |
>
> We observe that the DAC encoder is marginally better at preserving loudness in both the conditional generation and blending tasks, within the subband we condition on. The DAC encoder also produces similar audio quality for the blending task. However, the DAC encoder is worse at preserving timbral information, and the audio quality is worse for the conditional generation task. We hypothesize that this is because DAC’s reconstruction objective biases towards encoding low (audible) frequency information, while encoding less information about higher-frequency timbral information.
>
> [1] Hershey, Shawn, et al. "CNN architectures for large-scale audio classification." 2017 ieee international conference on acoustics, speech and signal processing (icassp). IEEE, 2017.
>
> [2] Amodei, Dario, et al. "Deep speech 2: End-to-end speech recognition in english and mandarin." International conference on machine learning. PMLR, 2016.
>
> [3] Choi, Keunwoo, George Fazekas, and Mark Sandler. "Automatic tagging using deep convolutional neural networks." arXiv preprint arXiv:1606.00298 (2016).
>
> [4] Kumar, Rithesh, et al. "High-fidelity audio compression with improved rvqgan." Advances in Neural Information Processing Systems 36 (2023): 27980-27993.

---

> ### Author Response · Authors · 2025-11-25
> **Official Comment by Authors (2/4)**
>
> **[W3 Understanding Ablations.]**
>
> We have added more analysis and explanation for each model component in the ablations section (B.1), along with supporting diagrams and figures.
>
> ---
> **[W3a. Necessity of Latent Augmentation.]**
>
> Applying frequency masking during training is essential to performance, since the decoder will see frequency-masked latents during inference. Without masking during training, the decoder does not learn how to reconstruct music from frequency-masked latents, and fails to generate high-quality audio during inference. (Table 9,10, under “w/o Frequency Masking” shows results without any latent-space augmentation).
>
> ---
> **[W3c. Log-Frequency.]**
>
> Log-frequency parameterization is ablated in the “w/o Log. Scale” experiment in Tables 9 and 10. We use a correlated mask to encourage the mask to form contiguous “groups”, instead of being speckled or erratic (justified further in Appendix B.1).
>
> However, most structured signals have a 1/f-spectrum, meaning that the energy at high frequencies is much smaller than the energy at low frequencies. Thus, a group of low-frequency bins contains much more energy than a group of high-frequency bins of equal width. To counterbalance this effect, we encourage high-frequency “groups” to be wider (see Figs. 7-8) by mapping the frequency bins to a logarithmic scale before computing correlations between bins. This reflects the fact that 1/f-spectra have equal energy per-octave.
>
> ---
> **[W4 Listening Study]**
>
> Now, we report statistics on the listening study in Appendix A.7. We include a Kruskal-Wallis H test, confirming that there are statistically significant pairs among the permutations. We also perform an analysis using the Wilcoxon signed rank test, which tests each pair of systems for statistically significant differences. We find that LatentFT outperforms all baselines in terms of audio quality to a significant extent, and all baselines in terms of “ability to blend” except for cross synthesis. We report p-values for each pairwise comparison in Table 8.
>
> We also calculate Fleiss’s Kappa to assess inter-rater reliability, which measures the degree of agreement beyond chance for multiple raters. We report κ = 0.0654 for our question about audio quality, and κ = 0.0914 for our question about “ability to blend”. Both values fall in the ”slight agreement” range [5], indicating substantial subjective variation in perceptual judgments. This is not uncommon in listening studies, where individual preferences naturally lead to varied responses.
>
> [5] Landis, J. Richard, and Gary G. Koch. "The measurement of observer agreement for categorical data." _biometrics_ (1977): 159-174.
>
> ---
> **[Q1 Failure Case Analysis]**
>
> We include failure cases for both conditional generation and blending on our website, under “Failure Cases” (https://latentfouriertransform.com/#failure_cases).
>
> In the first conditional generation failure case, we consider a clip with patterns at multiple timescales, but condition on a frequency band that contains no musical patterns (11-12 Hz). The resulting variation bears little resemblance to the original clip. We hypothesize that there is little information in the latent spectrum there to capture.
>
> In the second conditional generation failure case, we consider a clip with many sharp transients (bow strikes), and condition on very low frequency features (0-0.1 Hz). The resulting audio sounds like a smoothed version of the input clip, but has very low audio quality.
>
> In the first blending failure case, we choose the same latent frequency range for both references. This results in incoherent audio, indicating that the model is not yet able to compose features from overlapping latent frequency ranges.
>
> In the second blending failure case, we choose two adjacent latent frequency bands for the references. It also results in incoherent audio, showing that the model cannot combine musical patterns that occur at very similar timescales.
>
> ---
> **[Q5 Computational Costs of L]**
>
> For the DFT, we use PyTorch’s Fast-Fourier Transform (FFT) algorithm, which is $O(N  \cdot \log(N))$, where $N = L \cdot T’$ is the number of latent timeframes after zero padding. Thus, component is $O(L * log(L \cdot T’))$ (in both the forwards and backwards passes).
>
> Correlating the score matrix requires multiplying the $C \times F$ DFT spectrum by the correlation matrix $\mathbf{K} \in \mathbb{R}^{F \times F}$, which is an $O(C F^2)$ operation.
> Since $F = \text{floor}(LT'/2) + 1$, this is a $O(C (L \cdot T’)^2)$ operation (in both the forwards and backwards passes).
>
> Since our neural network contains many matrix multiplications (self attention, convolution, etc.), these computational costs are negligible during both training and inference. Thus, we choose L to be high enough to give a sufficient number of frequency bins, without worrying about the extra computational costs. In our experiments, L=2 (1024 bins) is enough, amounting to <0.2% of the total FLOP count for LatentFT-MLP.

---

> ### Author Response · Authors · 2025-11-25
> **Official Comment by Authors (3/4)**
>
> **[Clarification on DFT Orthogonality/W3b]**
>
> We thank the reviewer for raising a good point, and clarify the role of “DFT Orthogonality” below:
>
> First, since we are working with discrete-time signals, we consider the “DFT” of a signal to be synonymous with the “frequency-domain representation” of the signal.
>
> Thus, we consider our points about “DFT Orthogonality” to be **an argument for working in the frequency-domain generally**, and _not_ an argument for our specific latent-space augmentation (DFT masking).
>
> We summarize our arguments for why frequency-domain representations of the signal are useful:
> - In current deep multiscale representations, the fine representations of the signal and the coarse representations are entangled (often, one is derived from the other).
> - The frequency domain provides an alternative approach to multiscale representation,  since components of the signal at different scales are separated along a “frequency axis”.
> - Different frequency components of the signal are orthogonal to one another. This means that modifying the signal’s representation at one “scale” (frequency component) does not affect its representation at other scales.
>
> We thank the reviewer for the point about how DFT-masking may not be the only type of latent space that applies, and perform their suggested experiment in the section below:
>
> ---
> **[W2/Q4 Band-Drop Augmentation/W3b]**
>
> We thank the reviewer for their experiment suggestion! We had attempted and iterated on a similar augmentation before switching to DFT-Masking.
>
> Below, we show results replacing Latent DFT Masking with random latent bandpass augmentation. Instead of applying a DFT Mask to the latent space during training and inference, we apply a randomized bandpass filter to the latent space during training, and a user-specified one during inference.
>
> Results are shown in Appendix B.1, Tables 9 and 10 (under “w/ Bandpass Augmentation”).
>
> **Conditional Generation**
> | Method | Loud ↑ | Rhythm ↑ | Timbre ↓ | Harmony ↓ | FAD ↓ |
> |---|---|---|---|---|---|
> | **LatentFT-MLP** | 0.815 |  **0.963** | **0.376** | **0.079**      | **0.337** |
> | **w/ Bandpass Augmentation** | **0.861**  |   0.953    |  0.562    | 0.084    |   1.511     |
>
> **Blending**
> | Method | Loud ↑ | Rhythm ↑ | Timbre ↓ | Harmony ↓ | FAD ↓ |
> |---|---|---|---|---|---|
> | **LatentFT-MLP** | **0.686** |  0.873  | **1.021** | **0.108** |  **1.387** |
> | **w/ Bandpass Augmentation** |  0.664   |  **0.885**   |  1.636  | 0.117 | 2.58  |
>
> DFT-Masking has superior audio quality (lower FAD), while the metrics are somewhat similar in terms of adherence. Below, we contrast the two methods mathematically.
>
> **[Similarities to DFT Masking]**
>
> Although the bandpass augmentation does not explicitly compute the latent spectrum, it is quite similar to DFT masking. The bandpass filter attempts to select for the frequencies within its passband specification, while masking the DFT also attempts to select for the unmasked frequencies.
>
> Mathematically, convolution in the time domain is equivalent to multiplication in the spectral domain. Thus, applying a convolutional bandpass filter to the latent sequence is equivalent to multiplying the latent DFT spectrum by a (not 0-1) complex mask. In mathematical terms, the bandpass filter does the following:
>
> $$
> \text{Bandpass}(\mathbf{z}) = \text{Convolve}(\mathbf{z}, \mathbf{k}) = \text{DFT}(\mathbf{z}) \cdot \text{DFT}(\mathbf{k})
> $$
>
> While our DFT mask does the following:
> $$
> \text{DFT}(\mathbf{z}) \cdot \mathbf{M}
> $$
>
> Where $\mathbf{z}$ is the latent sequence, $\mathbf{k}$ is the convolutional kernel, and $\mathbf{M}$ is the DFT mask.
>
> There are two main difference between Bandpassing and DFT-Masking:
> DFT-Masking multiplies the spectrum by a 0-1 Mask, while bandpass filtering multiplies it by a softer mask with values outside of 0 and 1 (in terms of complex magnitude).
> The bandpass filter performs a linear convolution, while DFT masking is equivalent to a circular convolution.

---

> ### Author Response · Authors · 2025-11-25
> **Official Comment by Authors (4/4)**
>
> We continue our discussion of W2/Q4 Band-Drop Augmentation/W3b:
>
> **[Filter Lengths]**
>
> We switched to using the DFT for the following reason: FIR Bandpass filters use convolution, and require choosing a “filter length”. Short filters have a hard time isolating the lowest frequencies. For instance, bandpassing to 0 Hz is the same as reducing a signal to its average value, which means the filter must be as long as the signal itself.
>
> Meanwhile, long filters introduce substantial edge artifacts, due to the padding they require. For instance, if the filter is as long as the signal, the input signal must be padded to three times its length in order to preserve its shape post-convolution. This results in edge artifacts: For instance, if we use zero-padding, the result of the convolution will have a triangular shape, due to the effects of padding.
>
> DFT-Masking is equivalent to applying an ideal bandpass filter, where the input signal is padded periodically, and length of the filter equals the length of the signal. It is able to capture frequencies down to 0 Hz easily, since the 0th bin of $DFT(\mathbf{z})$ gives the average value. DFT-Masking also has an intuitive interpretation: It provides a least-squares approximation to the signal, using a sum of sinusoids corresponding to the unmasked frequencies.
>
> **[Training Stability/Orthogonality]**
>
> Bandpass filters are not ideal, meaning that they do not entirely reject the frequencies they try to filter out. Therefore, it should be noted that non-overlapping bandpass filters are generally **not orthogonal**, while non-overlapping DFT masks **are orthogonal**.
>
> We noticed that random bandpass augmentation suffered from training instability, requiring us to restart training twice with lower learning rates. We hypothesize that the orthogonality of the DFT contributes to training stability: in the backwards pass, the DFT mask can be interpreted as masking out orthogonal components of the upstream gradient. A more detailed explanation is in Appendix B.1.
>
>
> **[W2/"The most competitive baselines (ILVR, codec/spectrogram filtering) operate on spectrogram rather than latent frequency, creating an uneven comparison."]**
>
> We apologize for the confusion. To confirm and clarify, the ILVR, guidance, and spectrogram baselines perform frequency selection/masking on the frequency bins of the _DFT of the spectrogram_, not masking along the _frequency axis of the spectrogram_ (which correspond to audible frequencies). We clarify this in the paper (L397-399).
>
> Precisely, the object we are frequency masking is $\text{DFT}(\mathbf{x})$, where $\mathbf{x} \in \mathbb{R}^{80 \times 512}$ is a 5.9-second mel-spectrogram, and the DFT is taken along the spectrogram's time axis.
>
> The $80 \times 512$ mel-spectrogram has $C = 80$ channels (corresponding to audible frequency ranges) and $T' = 512$ timeframes.
>
> Similarly, our latent vector $\mathbf{z}$ also has $C = 80$ (latent) channels and $T' = 512$ timeframes. Thus, the number of spectrogram timeframes and the number of latent timeframes in LatentFT match each other, and so do the corresponding frame rates. Both DFTs are taken along the time axis of their respective representations.
>
> Thus, we view our baseline as comparing the $80 \times 512$ mel-spectral representation of audio against LatentFT's $80 \times 512$ latent representation, performing frequency masking on both. The DAC baseline applies the same concept to the  $1024 \times 512$ representation of the audio derived from the DAC encoder.

---

> ### Comment · Reviewer_YTEd · 2025-11-27
>
> I appreciate the authors' comprehensive response and the significant effort put into the additional experiments, particularly the DAC encoder comparison and the bandpass ablations. These additions, along with the statistical analysis of the user study, have clarified the technical mechanisms and improved the paper's rigor.
>
> However, I am maintaining my score of 4. The admitted infeasibility of scaling beyond 6-second clips significantly limits the contribution, restricting the method to local texture control rather than the broader claim of music generation involving structure or phrasing. Additionally, the performance drop with the DAC encoder suggests the approach is sensitive to the choice of representation, raising concerns about its robustness in modern raw-waveform workflows.

---

> ### Author Response · Authors · 2025-12-02
> **Response to Reviewer Response.**
>
> Thank you for your feedback. We address the two remaining points brought up by the reviewer.
>
> **[Scalability]**
>
> We now include 30-second examples on the website (https://latentfouriertransform.com/), addressing the reviewer's primary concern.
>
> Although training a 30-second model from scratch would have been infeasible given the rebuttal period and our computational budget, we realized that we can fine-tune our 5.9-second model on 30-second data from MTG-Jamendo:
> - The architecture (self-attention, convolutional layers) can expand to longer context lengths without raising an error.
> - The generation is locally conditioned on the masked latent $\mathbf{z}_\text{masked}$, which could suggest that a larger model architecture may not be needed for longer generations.
>
> We fine-tune our LatentFT model on 30s data for ~200k iterations, with a reduced batch size of 64 per GPU, adjusting the FFT length to 5120 (L=2) to account for the longer latent sequence length.
>
> The examples (https://latentfouriertransform.com/), show that the model generalizes to 30s after some fine-tuning. Also, we are able to condition on very low frequencies (0.05 Hz, or a period of 20 seconds), which are only able to be captured with longer generation lengths, and capture some sectional transitions.
>
> **[DAC Performance]**
>
> It should be noted that the LatentFT-DAC baseline required reducing the batch size from 256 to 64 during training (Appendix A.1), due to GPU memory constraints. This means that training LatentFT-DAC for the same number of iterations as the other LatentFT models resulted in fewer observed examples. This undertraining may have resulted in LatentFT-DAC's performance being slightly worse.
>
> To account for this, we will also report results LatentFT-DAC when it is trained for the _same number of data examples_ as the other LatentFT baselines. We will report these when they are ready.
>
> Also, it should be noted that although our reported LatentFT-DAC numbers were generally worse than the other LatentFT models, they are still generally better than the baselines (see Table 1). This implies that the encoder may be an important hyperparameter choice, but LatentFT is still a good framework.

---

### Official Review · Reviewer_jWrm · 2025-11-01

**Soundness:** 3
**Presentation:** 3
**Contribution:** 3
**Rating:** 6
**Confidence:** 4

**Summary:**

The paper introduces a generative audio framework that enables a novel frequency-domain control by manipulating musical patterns in the latent space according to their timescales. Their method:
- combines a diffusion autoencoder with a latent-space Fourier transform, trained end-to-end to decompose and control musical structures -- timescale directly in the latent space.
- Allows user-specified timescale conditioning, analogous to how equalizers shape timbre in audible frequencies. They use their model for generating music variation, blending, and separating patterns by timescale directly in the latent space with musically coherent results.
- Introduces new evaluation baselines tailored to assess this novel control task.

**Strengths:**

- The authors present a new method for guiding music generation by adjusting timescales in the latent space, offering an alternative to global, time varying, or token-based conditioning approaches.
- In the absence of prior studies on this type of control, they design thoughtful and relevant baselines, setting the stage for future developments in the field.
- The proposed method consistently outperforms existing baselines, demonstrating stronger conditioning behavior and clearer interpretability in blending and timescale separation tasks. Their analysis of the latent spectrum further reveals meaningful correlations with musical traits such as tempo, and pitch.
- The audio demonstrations show that their method is promising, and the upcoming code release will make the work easily reproducible and enhanced.
- Altogether, the paper lays a strong foundation for a new direction in generative audio research, opening the door to further refinements and creative extensions.

**Weaknesses:**

- The notion of timescale conditioning offers an interesting angle, but it provides a somewhat narrow view of musical structure. Many musical genres do not rely on clear, repeating patterns, which may limit the method’s generality beyond pattern-based music like pop or rock.
- It remains unclear whether the model can control acoustic attributes such as timbre or warmth, which reside in the actual frequency domain rather than in the latent frequency space. For instance, the diffusion model seems to receive no extra guidance, such as text or acoustic features, to generate the missing frequency content, resulting in more random variations and reduced controllability.
- Although the model can isolate latent frequencies, it struggles to disentangle higher-level musical concepts such as instruments or timbral components. This raises concerns about scalability, for instance, how would it handle evolving patterns or instruments in a full-length piece, given that training and evaluation rely on short (≈5.9 s) audio clips?
- The audio quality of the generated samples is still limited, with noticeable metallic artifacts that affect realism and listening comfort.

**Questions:**

- Line 352: The author state that in the Guidance and ILVR baselines, the diffusion process is guided using Mel spectrograms rather than the latent spectrum. Why was this approach chosen, and wouldn’t applying the DFT directly to the latent space make for a more comparable baseline?
- Could the authors clarify the influence of framerate T’ on the results? How should it be chosen to effectively capture musical patterns?
- Minor note: in line 830, there appears to be an extra space that should be corrected.

---

> ### Author Response · Authors · 2025-11-25
> **Official Comment by Authors (1/2)**
>
> We thank the reviewer for their valuable feedback. We agree that our work lays a strong foundation for a new direction in generative audio research.
>
> ---
>
> **[Clear, Repeating Patterns]**
>
> The reviewer raises a good point.
>
> We find that the music does not necessarily have to contain clear, repeating patterns in order to apply our framework. We would like to highlight the example under “0 - 0.5 Hz example” in our demo website (https://latentfouriertransform.com/#cond, “Conditional Generation”), where we condition several musical variations on a non-repeating pattern captured from some classical guitar arpeggios.
>
> We also show new examples under “Changes Between Sections” (https://latentfouriertransform.com/#transitions), where the song goes from one section to another, and we are able to generate effective variations. We believe that this implies to some extent that the patterns in the music can change, and LatentFT will still work.
>
> We also show new examples on the Maestro dataset (piano music) [1] on the website (https://latentfouriertransform.com/#other_styles) which is not pop or rock related. We provide quantitative results on this dataset in Table 12 in Appendix B.2.
>
> ---
>
> **[Controlling other Attributes]**
>
> Although our work does not directly control attributes like timbre and warmth, we believe that control along the ‘timescale’ axis can complement other forms of control, like text or acoustic features. For instance, a user may want to condition on the loudness, pitch, or timbral components from a reference, but at a specific timescale.
>
> Our form of conditioning can be combined with other types of conditioning, like text-based conditions, timbre-based conditions, or conditions based on acoustic features. For instance, we could train our diffusion autoencoder with text conditioning, by conditioning the diffusion model on both the text encoding obtained from the caption (via cross attention), as well as the latent feature vector obtained from the input audio.
>
> The ideal system combines control along both semantic axes and timescale axes. We focus specifically on demonstrating our unique, frequency-based method of control, since it is an orthogonal axis that provides a new form of control to music generation models.
>
> ---
>
> **[Evolving Patterns]**
>
> While we have shown the usefulness of LatentFT on capturing chord changes, rhythmic and melodies, the original paper did not address our framework’s usefulness on larger-scale musical phenomena like evolving patterns. As previously mentioned, our new examples show how LatentFT handles changes between sections (https://latentfouriertransform.com/#transitions) and indication that LatentFT can deal with changing musical patterns.
>
> Ideally, showing that we could capture evolving patterns would involve working with longer segments (30s - 3 min., instead of 5.9s), which we could not do due to computational constraints. However, we have reason to believe that the model’s conditioning will still work even if the patterns at the selected timescales change. Conditioning on a high-frequency pattern, for instance, essentially involves applying a high-pass filter to the latent sequence. If the song has multiple sections, each with different rhythmic patterns, we hypothesize that this would retain the fast-moving patterns from each section. We acknowledge that we cannot say for sure what would happen on longer clips, and that is a promising direction for future work.
>
>
> ---
>
>
> **[Audio Quality]**
>
> Although the audio quality can be improved, we maintain that it is still fairly listenable, and effectively demonstrates our approach. We encourage all readers to go to https://latentfouriertransform.com/ to evaluate the audio quality of the examples.

---

> ### Author Response · Authors · 2025-11-25
> **Official Comment by Authors (2/2)**
>
> **[Latent Space DFT for Baselines?]**
>
> We appreciate the request for clarification with regards to applying the DFT to the latent space for the Guidance and ILVR baselines:
>
> The Guidance and ILVR baselines do not actually have a latent space the same way our model does, since they are based on unconditional diffusion models that generate mel-spectrograms. In both baselines, we initiate an unconditional diffusion process that we guide to be similar to our reference at the selected timescales.
>
> Specifically, the guidance baseline computes a loss between  $\text{DFTMASK}(\mathbf{x}_0)$ and $\text{DFTMASK}(\hat{\mathbf{x}}_0)$, where $\mathbf{x}_0$ is the reference mel-spectrogram, and $\hat{\mathbf{x}}_0$  is the diffusion model’s estimate of the clean mel-spectrogram during the denoising process. This loss is used to guide the denoising trajectory.
>
> Note that we are taking the DFT of a _Mel-spectrogram_ along its time axis. This is because the DFT mask selects very low frequencies, between 0 - 43 Hz. These frequencies in audio-space are not meaningful, since they are below the audible spectrum. So, we select the DFT of the Mel-spectrogram (resulting in a type of modulation spectrum) as an alternative representation.
>
> Tthe ILVR baseline essentially _replaces_ selected frequency components of $\text{DFT}(\hat{\mathbf{x}}_0)$ with those of $\text{DFT}(\mathbf{x}_0)$ at each denoising step. Again, we are replacing frequency components of the DFT of the mel-spectrogram.
>
> ---
>
> **[Framerate T’]**
>
> To first clarify, T’ is actually the number of latent timeframes. For instance, in our model, a 5.9 second clip is encoded into T’=512 latent timeframes. T’ is inversely proportional to the latent framerate $f_r$, which is the number of latent timeframes per second.
>
> The latent framerate determines the maximum frequency that the latent spectrum can capture (the Nyquist frequency), which is $f_r / 2$. As the latent framerate gets slower, the highest frequency that the latent spectrum can represent decreases.
>
> Thus, if we choose the latent framerate too low, we will not be able to capture rapid, high-frequency patterns. On the other hand, lower framerates may allow the encoder to aggregate more semantic information into each latent frame, resulting in a more powerful representation.
>
> To balance these two, we suggest choosing the latent framerate to be around 40-80 Hz, resulting in a latent Nyquist frequency between 20-40 Hz. Choosing a higher latent framerate would not benefit us, since structures in the audio signal above this frequency are perceived as pitch, instead of musical structure. Choosing a lower framerate than this may result in better aggregation of semantic information, but could begin to set a restrictive limit on the range of the latent spectrum.
>
> For our experiments, $f_r \approx 86$ Hz, giving a latent Nyquist of 43 Hz. This corresponds to a downsampling rate of 256 from the waveform, which is a convenient power-of-two downsampling factor for deep learning architectures.
>
> ---
>
> **[Extra Space]**
>
> Thank you for the correction!

---

### Meta-Review · Area_Chair_usTg · 2026-01-07

**Summary:**

The positive reviews appreciate the idea that latent-time Fourier coordinates plus frequency-masked training yields a controllable, interpretable timescale axis that is meaningfully different from existing representations or use of audible-frequency filtering and other conditioning. The borderline/negative reviews worry that the paper’s claims outrun the evidence because the original experiments were mostly on short (~6s) clips, and questioning whether improvements come from Fourier analysis or some other latent augmentation aspect. There was uncertainty about interpretability and consistency across songs/styles and whether the approach is sensitive to encoder choice.

**Reviewer Concerns:**

The rebuttal addressed most methodological concerns by clarifying the definition of latent frequency as a representation and adding ablations and showing additional experiments that show that removing the latent DFT or omitting masking during training leads to failures. Testing on additional datasets, alternative encoders, longer examples, and explicit failure cases suggests the idea is pretty robust and probably extends beyond the narrow original setup. This seems to be addressing most of the reviewer concerns.  Issue of scale still remain and while longer examples help, they do not yet fully establish stability or interpretability at truly long musical horizons.

**Reviewer Scores:**

After rebuttal, the paper seem to move up from split reject/accept to borderline-accept, with two clear supporters, one rather skeptic review and one likely remaining as a reject. Post rebuttal the remaining objections seem to be about overstating the importance and scope of the contribution rather then novelty or technical validity, which is probably ok for papers that introduce a new representational idea rather than a fully mature end-to-end system.

---

### Decision · Program_Chairs · 2026-01-26

Accept (Oral)